# Assembling covalent organic framework membranes with superior ion exchange capacity

Xiaoyao Wang [1,2,7], Benbing Shi[1,2,7], Hao Yang [3], Jingyuan Guan[1,2], Xu Liang[1,2], Chunyang Fan[1,2], Xinda You [1,2], Yanan Wang[1,2], Zhe Zhang[3], Hong Wu [1,2,4], Tao Cheng[3], Runnan Zhang [1,2,5 ✉] & Zhongyi Jiang [1,2,5,6 ✉]

Ionic covalent organic framework membranes (iCOFMs) hold great promise in ion conduction-relevant applications because the high content and monodispersed ionic groups could afford superior ion conduction. The key to push the upper limit of ion conductivity is to maximize the ion exchange capacity (IEC). Here, we explore iCOFMs with a superhigh ion exchange capacity of 4.6 mmol g$^{-1}$, using a dual-activation interfacial polymerization strategy. Fukui function is employed as a descriptor of monomer reactivity. We use Brønsted acid to activate aldehyde monomers in organic phase and Brønsted base to activate ionic amine monomers in water phase. After the dual-activation, the reaction between aldehyde monomer and amine monomer at the water-organic interface is significantly accelerated, leading to iCOFMs with high crystallinity. The resultant iCOFMs display a prominent proton conductivity up to 0.66 S cm$^{-1}$, holding great promise in ion transport and ionic separation applications.

[1] Key Laboratory for Green Chemical Technology of Ministry of Education, School of Chemical Engineering and Technology, Tianjin University, 300072 Tianjin, China. [2] Collaborative Innovation Center of Chemical Science and Engineering (Tianjin), 300072 Tianjin, China. [3] Institute of Functional Nano&Soft Materials (FUNSOM), Jiangsu Key Laboratory for Carbon-Based Functional Materials & Devices, Soochow University, 215123 Suzhou, China. [4] Tianjin Key Laboratory of Membrane Science and Desalination Technology, Tianjin University, 300072 Tianjin, China. [5] Zhejiang Institute of Tianjin University, 315201 Ningbo, Zhejiang, China. [6] Joint School of National University of Singapore and Tianjin University, International Campus of Tianjin University, 350207 Binhai New City, Fuzhou, China. [7] These authors contributed equally: Xiaoyao Wang, Benbing Shi. ✉email: runnan.zhang@tju.edu.cn; zhyjiang@tju.edu.cn

C ovalent organic frameworks (COFs) have evolved into a class of most promising emergent porous membrane materials owing to their high surface area, tunable pore sizes, predictable pore apertures, easy functionalization and superior hydrothermal stability[1–11]. For various applications such as gradient energy conversion, ion conduction[3,4,12–15], ion separation[16] and molecular sieving[17], ionic COF membranes (iCOFMs) hold great potential because the high content and monodispersed ionic groups could afford superior ion conductivity[10].

Increasing ion exchange capacity (IEC) of membranes by introducing multiple ion groups into skeletons is the key to push the upper limit of iCOFM proton conductivity[3,10]. Compared to post-synthetic strategies, De Novo strategies by designing ionic monomers can guarantee the all-dimensional and homogeneous modification inside the COF membrane and has been proved effective for iCOFMs design[3,4,18]. Schiff base reaction between amine monomer and aldehyde monomer is most frequently utilized for COF membrane fabrication[3,4,13,16,18–36], owing to the rich monomer designability and mild reaction conditions[1,22,37]. Among the monomers for Schiff-base reactions, the ionic amine monomers are the essential building blocks for iCOFMs[6,12–15,38,39]. Benefited from the simplicity and scalability, interfacial polymerization (IP) has evolved as a platform technology for COF membrane fabrication by confining the polymerization reactions between monomers in two immiscible phases at the interface[19]. During the IP process, the monomer reactivity directly governs the membrane structure formation[24,31]. However, the IP technology for fabricating non-ionic COF membranes can hardly be transplanted into iCOFMs fabrication directly. This is because the monomers with multiple ionic groups usually exhibit extremely low reactivity due to the strong electron-withdrawing and steric hindrance effect of ionic groups[10]. Moreover, the protons of ionic groups or acid activators (catalysts) can easily combine with –NH$_2$ to form –NH$_3^+$, which will inhibit the nucleophilic attack ability of the amine monomer[19,22,40]. Fukui function, first proposed by Parr and Yang in 1984, has been widely applied as a tool for deducing the relative reactivity of different positions in a molecule[41]. In this regard, we employed the Fukui function as a descriptor to compare the reactivity of amine monomers with different ionic group numbers, (a. p-phenylenediamine with no ionic group, b. 2,5-diaminobenzenesulfonic acid with one ionic group, c. 4,4′-diaminobiphenyl-3,3′-disulphonic acid with two ionic group). We found that the Fukui function for electrophilic attack sites of three types of monomers was a(0.123) > b(0.017) >c(0.009), indicating that the reactivity of amine monomer decreased as the number of ionic groups increased. Therefore, we speculate that appropriate activation of ionic monomers should be the key to fabricating iCOFMs with high IEC by using IP technology.

In this work, we explore TpBD-(SO$_3$H)$_2$ iCOFMs with superhigh IEC of 4.6 mmol g$^{-1}$ for proton conduction, using our dual-activation interfacial polymerization strategy. We use Brønsted acid to activate aldehyde monomers in organic phase and Brønsted base to activate ionic amine monomers in water phase. Brønsted acid can convert –CHO into cation –C$^+$HOH, which is more liable to nucleophilic attack[42], and accordingly the Fukui function for nucleophilic attack sites of aldehyde monomer increases from 0.066 to 0.152. Brønsted base can convert –NH$_3^+$ into more active nucleophile –NH$_2$ to achieve the activation of amine monomers, and accordingly the Fukui function for electrophilic attack sites of amine monomers increases from 0.009 to 0.072. After dual-activation, the Schiff-base reaction at the water–organic interface is accelerated significantly, leading to robust and highly crystalline iCOFMs. In sharp contrast, we cannot obtain crystalline iCOFMs by single-activation (only activating one monomer) or without activation. Owing to the superhigh IEC and highly continuous ionic channels, our De Novo designed TpBD-(SO$_3$H)$_2$ iCOFMs exhibits superior proton conductivity of 0.66 S cm$^{-1}$ (90 °C, 100% relative humidity).

## Results

The Schiff base-linked TpBD-(SO$_3$H)$_2$ iCOFMs were prepared by the dual-activation interfacial polymerization strategy, as illustrated in Fig. 1a. Ionic amine monomer 4,4′-diaminobiphenyl-3,3′-disulfonic acid (BD-(SO$_3$H)$_2$) in water was activated by sodium formate (a Brønsted base), while the aldehyde monomer 2,4,6-triformylphloroglucinol (Tp) in organic phase was activated by n-octanoic acid (a Brønsted acid). When the interfacial polymerization started, the activated BD-(SO$_3$H)$_2$ monomers diffused into the organic phase and reacted with the activated Tp monomers at the water–organic interface. After dual-activation, the higher reaction rate could generate COF particles instead of continuous COF membranes[31]. As the reaction time prolonged, the COF particles will assemble into a loose layer, which reduced the diffusion rate of amine monomers into the organic phase. Accordingly, the rate of interfacial polymerization slowed down, leading to the formation of a compacted layer on the top side. After 24 h, defect-free TpBD-(SO$_3$H)$_2$ iCOFMs formed, which exhibited an asymmetrical structure (Supplementary Fig. 1). The SEM and AFM images of the TpBD-(SO$_3$H)$_2$ iCOFMs toward organic phase side showed a smooth and defect-free membrane surface with a roughness of 8.20 nm (Fig. 1b). The TpBD-(SO$_3$H)$_2$ iCOFMs toward water phase side showed loose membrane structure (Supplementary Fig. 1). The TpBD-(SO$_3$H)$_2$ iCOFMs displayed mechanical strength of 20 MPa, ensuring great potential in practical applications. (Supplementary Fig. 2). The chemical composition of the TpBD-(SO$_3$H)$_2$ iCOFMs was analyzed by FTIR (Supplementary Fig. 3) and solid-state NMR spectroscopy (Fig. 1c). The characteristic frequencies at 1565–1576 cm$^{-1}$ (C=C stretching vibration), and 1267–1292 cm$^{-1}$ (C–N stretching vibration) from the FTIR spectra and the characteristic peaks at ~184.8 ppm (C=O), ~101.9 ppm (C=C) and 147.0 ppm (C–N) of the $^{13}$C solid-state NMR spectra verified the formation of the β-ketoenamine framework structure[37].

The crystallinity of the TpBD-(SO$_3$H)$_2$ iCOFMs was probed by the PXRD technique (Fig. 2a). The experimental PXRD pattern was highly consistent with the simulated reversed AA stacking PXRD patterns with the existence of the sharp diffraction peaks corresponding to the 100 planes. Interestingly, the TpBD-(SO$_3$H)$_2$ iCOFMs fabricated by dual-activation interfacial polymerization at room temperature exhibited higher crystallinity compared with the same type of COF powder synthesized by solvothermal method at 120 °C (Supplementary Fig. 4). Meanwhile, we conducted the grazing incidence wide-angle X-ray scattering (GIWAXS) and grazing incidence XRD (GIXRD) measurement to analyze the crystallinity of the membrane surface. The Bragg peaks appeared as diffraction rings in Fig. 2b, indicating the high membrane surface crystallinity. It was found that the crystallinity of the top dense side of the membrane was slightly higher than that of the loosely stacked side (Fig. 2b and Supplementary Figs. 5 and 6). The diffraction ring of 2.4 nm$^{-1}$ represented the 100 reflection plane of TpBD-(SO$_3$H)$_2$, which was well consistent with the PXRD pattern of the TpBD-(SO$_3$H)$_2$ iCOFMs. Moreover, the high-resolution transmission electron microscopy (HRTEM) images of the slices of TpBD-(SO$_3$H)$_2$ iCOFMs verified the ordered crystalline structures of our membrane, and the EDS image of the membrane slices proved the monodispersed distribution of sulfonic acid groups in the membrane (Fig. 2c). To obtain the porosity, we performed N$_2$ and water vapor adsorption analysis of the TpBD-(SO$_3$H)$_2$ iCOFMs at 77 K (Fig. 2d and Supplementary Fig. 7). The

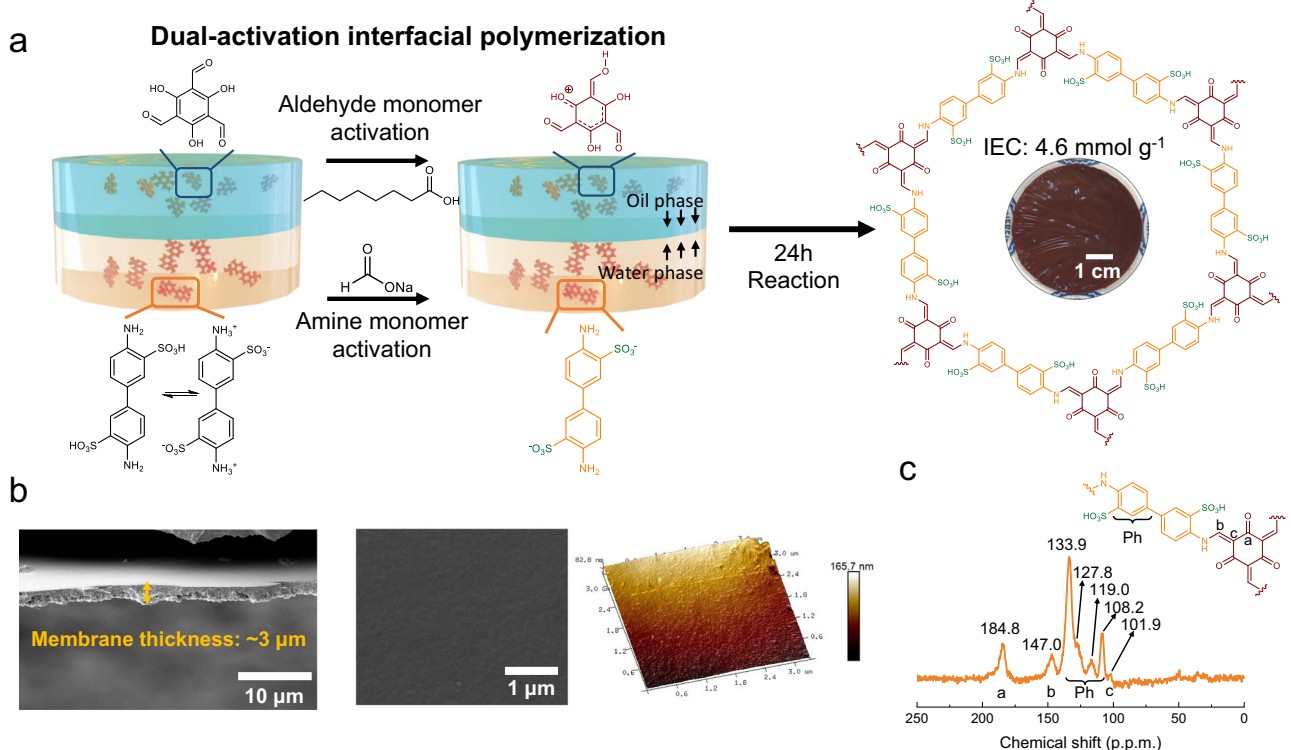

**Fig. 1 Membrane fabrication. a** Schematic illustration of the TpBD-(SO$_3$H)$_2$ iCOFMs fabrication process. **b** Cross-sectional SEM image, surface SEM image and surface AFM images of the TpBD-(SO$_3$H)$_2$ iCOFMs. **c** Solid-state $^{13}$C NMR spectra of the TpBD-(SO$_3$H)$_2$ iCOFMs.

Brunauer–Emmett–Teller (BET) surface area of the TpBD-(SO$_3$H)$_2$ iCOFMs was 203.4 m$^2$ g$^{-1}$, and the water vapor adsorption capacity was 477 mg g$^{-1}$. The TGA result revealed that the TpBD-(SO$_3$H)$_2$ iCOFMs was stable below 300 °C, which meets the requirement of practical application (Supplementary Fig. 9). Notably, no obvious change in the PXRD patterns of the TpBD-(SO$_3$H)$_2$ iCOFMs could be observed after immersion in 3 M H$_2$SO$_4$, 3 M NaOH solution and N,N-dimethylformamide (100 °C) for 5 days (Supplementary Fig. 10), indicating the high-structural stability of the TpBD-(SO$_3$H)$_2$ iCOFMs skeleton due to the irreversible enol-to-keto tautomerization[37,43]. To elucidate the effect of the monomer activation on the TpBD-(SO$_3$H)$_2$ iCOFMs growth, the IP process was carried out under the following cases: non-activation, single-activation (aldehyde monomer activation or amine monomer activation), and dual-activation. It can be seen that no iCOFMs formed at the water–organic interface under non-activation condition even after 72 h (Fig. 3a). Moreover, no Tyndall effect was observed in the water phase, indicating that no reaction took place between these two monomers. Under the single aldehyde monomer activation condition, a weak Tyndall phenomenon appeared in the aqueous solution, suggesting that the reaction between the amine monomer and the aldehyde monomer has occurred, and COF nanosheets were formed in the water phase (Fig. 3b). However, the reaction speed was too slow for iCOFMs growth and formation. Under the single amine monomer activation condition, the TpBD-(SO$_3$H)$_2$ iCOFMs formed with a thickness of only 4 μm (Fig. 3c), and the membrane mechanical strength was too weak to be used in the subsequent experiments. As shown in the membrane PXRD pattern (Fig. 3c), the weak 100 characteristic peak indicates low crystallinity of iCOFMs. Under the dual-activation condition (Fig. 3d), the TpBD-(SO$_3$H)$_2$ iCOFMs with a thickness of 85 μm was formed, which exhibited an intense 100 peak in the PXRD pattern, revealing the high crystallinity of the membranes.

Based on the above experimental results, we propose the tentative mechanism of the dual-activation interfacial polymerization. As shown in Fig. 3e, f, without activation, the Fukui function $f^+$ for nucleophilic attack sites on the aldehyde monomer was 0.066. After the aldehyde monomer was dissolved in Brønsted acid, the oxygen atoms in the aldehyde group could capture the protons of octanoic acid to form –C$^+$HOH. In particular, the –C$^+$HOH of Tp monomer could undergo keto–enol tautomerization, placing the positive charge on the –OH group (Fig. 3f)[42]. After tautomerization, the carbon atom of aldehyde group becomes more sensitive to nucleophilic attack. The Fukui function $f^+$ of carbonyl carbon atom on Tp molecule increased to 0.152, indicating the increased monomer reactivity. For amine monomers, it can be seen from the optimized BD-(SO$_3$H)$_2$ structure (calculated by the Gaussian 16 program, Supplementary Fig. 11), the sulfonic acid groups and amine groups in BD-(SO$_3$H)$_2$ monomers could form intramolecular hydrogen bonds. And because the amine groups have more negative charges (Supplementary Fig. 11), the proton on the sulfonic acid group would be closer to the amine group to occupy the lone pair of electrons of the N atom, rendering it more difficult for nucleophilic attack of the amine group. The calculation results showed that Fukui function for electrophilic attack sites $f^-$ of ionic amine monomer BD-(SO$_3$H)$_2$ was 0.009, which was much lower than that of the non-ionic amine monomers ($f^-$ value above 0.07) (Supplementary Tables 2 and 3). The huge gap between the reactivity of ionic amine monomers and non-ionic amine monomers can explain the difficulty to fabricated the TpBD-(SO$_3$H)$_2$ iCOFMs by the IP technology with non-activation or single-activation (aldehyde monomer activation). After activation, the proton that may inhibit the nucleophilic attack of the ionic amine monomer was taken by the Brønsted base. The $f$ value increased from 0.009 to 0.072, a level similar to that of non-ionic amine monomers (Fig. 3f and Supplementary Table 4). However, under single-activation condition (amine monomer

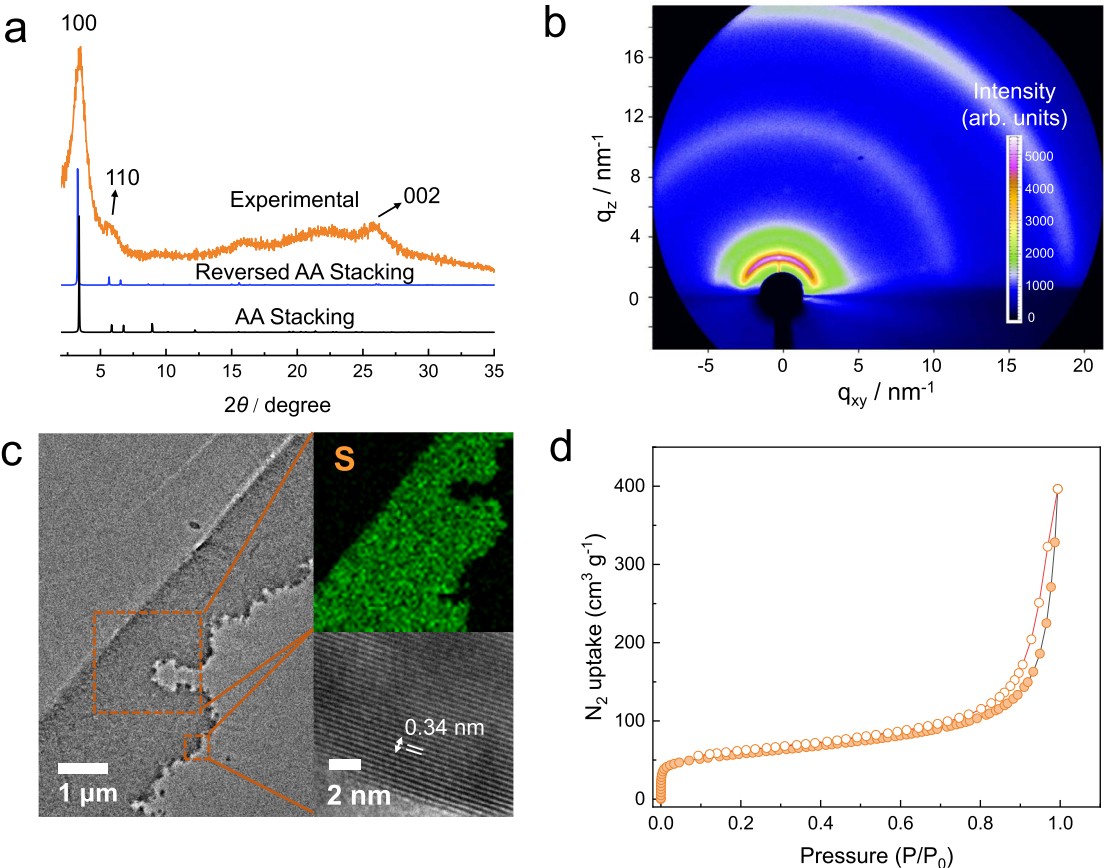

**Fig. 2 Membrane characterization. a** Experimental PXRD pattern, and simulated PXRD patterns for AA and reversed AA stacking of the TpBD-(SO₃H)₂ iCOFMs. **b** GIWAXS pattern of the TpBD-(SO₃H)₂ iCOFMs. **c** TEM, EDS, and high-resolution TEM images of the slices of the TpBD-(SO₃H)₂ iCOFMs. **d** N₂ sorption isotherms of the TpBD-(SO₃H)₂ iCOFMs.

activation), because the aldehyde monomer was not activated, the relatively high reaction energy barrier could affect the long-range order of the framework and result in iCOFMs with poor crystallinity[40]. Finally, after dual-activation, the Schiff-base reaction at the water–organic interface was significantly accelerated, leading to robust and highly crystalline iCOFMs.

According to the proposed dual-activation interfacial polymerization mechanism, other similar Brønsted acids and bases also could be used for monomer activation. To explore the selection criteria of activators, we further selected a variety of amine monomer activators (sodium acetate and sodium benzoate) and aldehyde monomer activators (n-heptanoic acid, n-nonanoic acid and n-decanoic acid) for the TpBD-(SO₃H)₂ iCOFMs fabrication. As shown in Supplementary Figs. 13 and 14, a series of iCOFMs can be obtained with different thickness and crystallinity. We found that the membrane thickness increased with the increase of the acidity of the aldehyde monomer activator. This phenomenon can be explained by the membrane growth mechanism. As the acidity of the aldehyde monomer activator increased, aldehyde monomers were more easily to be activated and the initial interfacial polymerization rate became higher, leading to the increased thickness of the loose layer of the TpBD-(SO₃H)₂ iCOFMs. Similarly, as the basicity of the amine monomer activator increased, the membrane thickness increased. In addition, the TpBD-(SO₃H)₂ iCOFMs obtained by coupling sodium formate and n-octanoic acid exhibited highest crystallinity (Supplementary Fig. 14). Therefore, sodium formate and n-octanoic acid were selected as model activators for further study.

Proton conduction is an essential property of iCOFMs. IEC, thickness, crystallinity, and ion conduction channel continuity of iCOFMs play a decisive role in the proton conductivity. In this study, the membrane thickness, crystallinity, and channel continuity can be adjusted by changing the addition amount of monomer activators. As shown in Fig. 4a, with the increasing amount of n-octanoic acid in the organic phase, the thickness of membrane loose layer increased, while the thickness of the dense layer remained almost unchanged (~3 μm, Supplementary Fig. 15). This phenomenon can be explained by the membrane growth mechanism. With the increase of the n-octanoic acid addition amount, more aldehyde monomers were activated and the initial interface polymerization rate became higher, leading to the formation of more non-continuous COF particles and, accordingly, the increase of the thickness of the membrane loose layer. As the n-octanoic acid addition amount increased from 5 to 15 mL, the 100 characteristic peak intensity was enhanced (Fig. 4b) and the proton conductivity of the membrane increased from 0.33 to 0.66 S cm⁻¹. The enhanced proton conductivity was attributed to the ordered and continuous ionic channels formed in the highly crystalline TpBD-(SO₃H)₂ iCOFMs, which can effectively facilitate the proton transport[3]. However, the proton conductivity decreased from 0.66 to 0.13 S cm⁻¹ with the increase of n-octanoic acid amount from 15 to 20 mL (Fig. 4c). This was due to the increased thickness of the low-conductive loose layer of the TpBD-(SO₃H)₂ iCOFMs, within which the ionic channels were less continuous. Similarly, the membrane thickness increased with the increase of sodium formate addition amount (Fig. 4d). With low sodium formate addition amount (0.5–1eq),

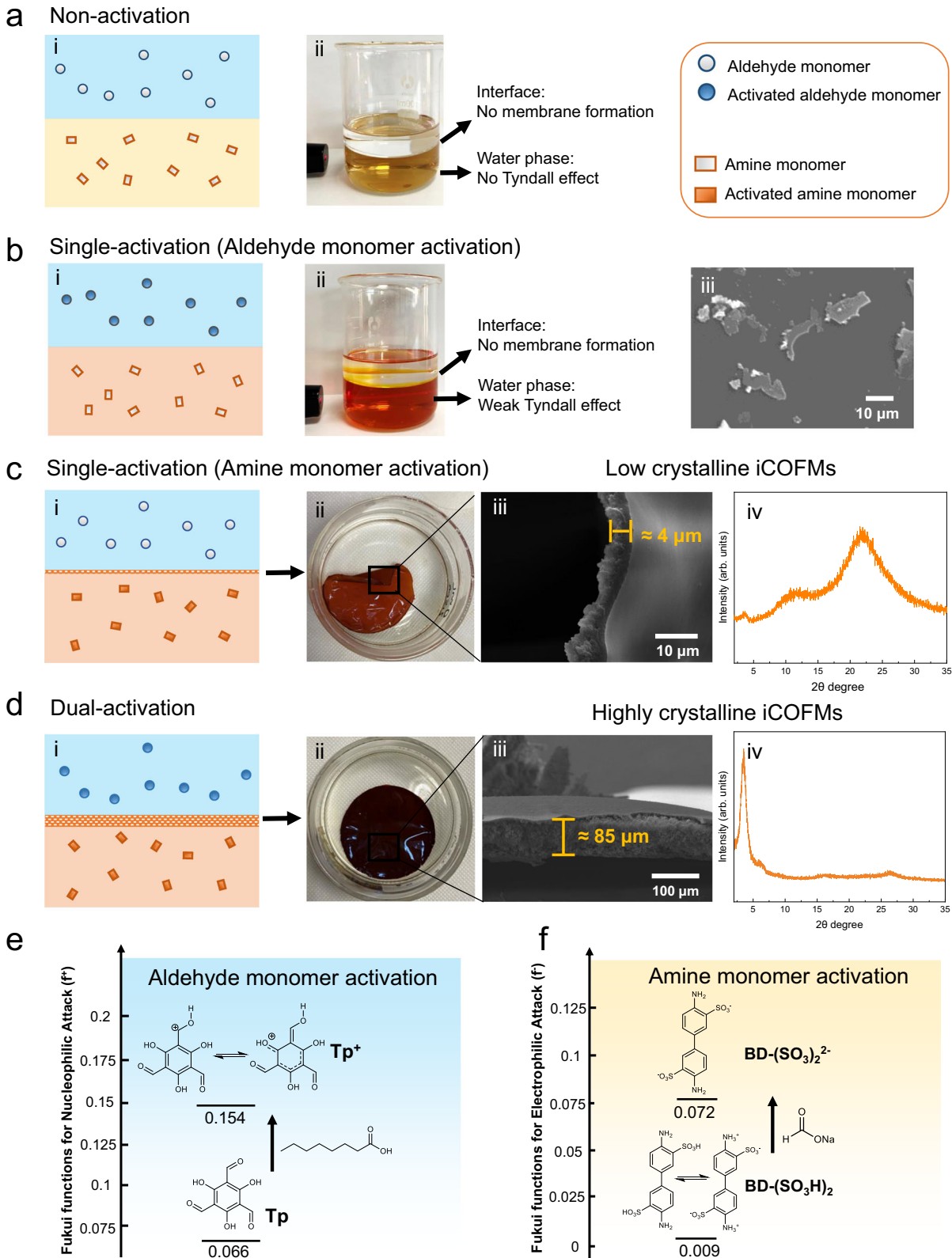

**Fig. 3 Mechanism of dual-activation interfacial polymerization. a** Fabrication of the TpBD-(SO$_3$H)$_2$ iCOFMs by non-activation IP: (i) schematic diagram and (ii) digital photo of the IP system after 72 h of reaction. **b** Fabrication of the TpBD-(SO$_3$H)$_2$ iCOFMs by single-activation IP (aldehyde monomer activation): (i) schematic diagram, (ii) digital photo of the IP system after 72 h of reaction, and (iii) SEM image of the TpBD-(SO$_3$H)$_2$ COF nanosheets in the water phase. **c** Fabrication of the TpBD-(SO$_3$H)$_2$ iCOFMs by single-activation IP (amine monomer activation): (i) schematic diagram, (ii–iv) digital photo, cross-section SEM image and PXRD pattern of the TpBD-(SO$_3$H)$_2$ iCOFMs. **d** Fabrication of the TpBD-(SO$_3$H)$_2$ iCOFMs by dual-activation IP: (i) schematic diagram, (ii–iv) digital photo, cross-section SEM image and PXRD pattern of the TpBD-(SO$_3$H)$_2$ iCOFMs. **e** Fukui functions for nucleophilic attack sites ($f^+$) of aldehyde monomer. **f** Fukui functions for electrophilic attack sites ($f^-$) of amine monomer.

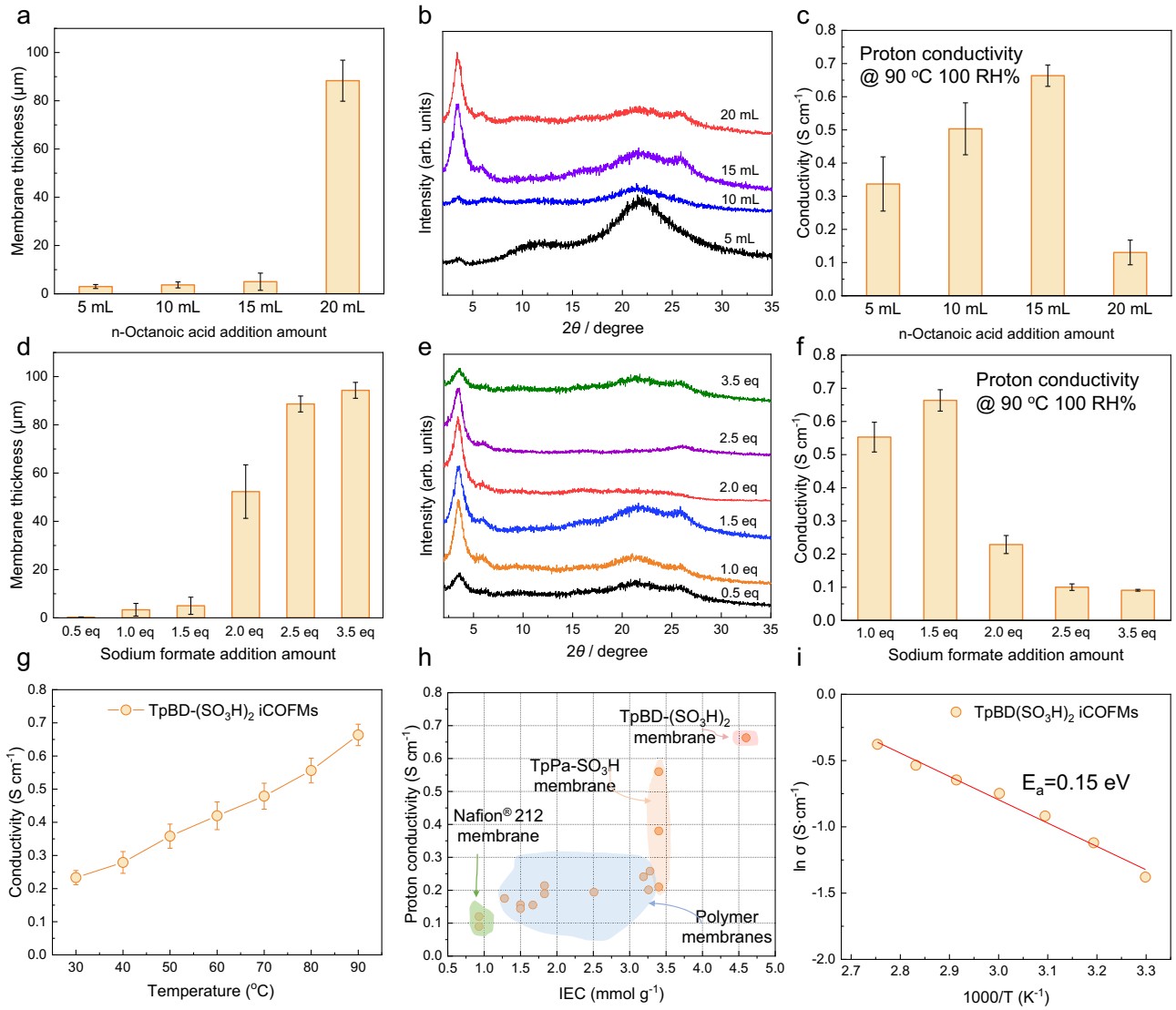

**Fig. 4 Membrane structure manipulation and proton conductivity evaluation. a–c** Effect of the n-octanoic acid addition amount on (**a**) thickness, (**b**) crystallinity, and (**c**) proton conduction performances of the TpBD-(SO$_3$H)$_2$ iCOFMs. Curves in (**b**) are the PXRD patterns of TpBD-(SO$_3$H)$_2$ iCOFMs fabricated with the n-octanoic acid addition amount of 5 mL (black), 10 mL (blue), 15 mL (purple), and 20 mL (red). In this experiment, 1.5 equivalents (eq) of sodium formate were added to the water phase as the amine monomer activator. **d–f** Effect of the sodium formate addition amount on (**d**) thickness, (**e**) crystallinity and (**f**) proton conduction performances of the TpBD-(SO$_3$H)$_2$ iCOFMs. Curves in (**e**) are the PXRD patterns of TpBD-(SO$_3$H)$_2$ iCOFMs fabricated with the sodium formate addition amount of 0.5 eq (black), 1.0 eq (orange), 1.5 eq (blue), 2.0 eq (red), 2.5 eq (purple), and 3.5 eq (green). In this experiment, 15 mL of n-octanoic acid were added to the organic phase as the aldehyde monomer activator. **g** Temperature-dependent proton conductivity of the TpBD-(SO$_3$H)$_2$ iCOFMs fabricated at optimized condition (the addition amount of n-octanoic acid and sodium formate was 15 mL and 1.5 eq, respectively). **h** IEC value versus proton conductivity for polymer membranes, previously reported iCOFMs and the TpBD-(SO$_3$H)$_2$ iCOFMs (detailed data was provided in Supplementary Table 9). **i** Proton-transfer activation energy ($E_a$) of the TpBD-(SO$_3$H)$_2$ iCOFMs fabricated at optimized condition. All the error bars in this figure represent the standard deviation ($n = 3$ independent experiments), data are presented as mean values ± SD.

the resultant TpBD-(SO$_3$H)$_2$ iCOFMs was relatively thin (~0.2–2 μm), due to the low interfacial polymerization rate. In this case, the mechanical strength of the TpBD-(SO$_3$H)$_2$ iCOFMs was insufficient for the proton conduction test. When the addition amount of sodium formate increased from 0.5 to 2.5 eq, the crystallinity of the TpBD-(SO$_3$H)$_2$ iCOFMs was enhanced (Fig. 4e). However, when the addition amount of sodium formate further increased from 2.5 to 3.5 eq, the characteristic peak intensity turned lower. This was because the excessive interfacial polymerization rate will lead to rapid formation of low-crystalline COF particles. Accordingly, the crystallinity of the loose layer became lowered. Similar phenomena have also been observed in previous studies[31,44]. When the addition amount of sodium

formate was 1.5 eq and the resultant TpBD-(SO$_3$H)$_2$ iCOFMs possessed a thinner loose layer and high crystallinity. Under this optimized condition, the resultant TpBD-(SO$_3$H)$_2$ iCOFMs exhibited superior proton conductivity (0.66 S cm$^{-1}$ at 90 °C and 100% relative humidity (RH), Fig. 4f), which was remarkably higher than those of reported COF-based proton-conducting materials, and was nine-fold higher than the commercial Nafion 212 membrane[3] (Supplementary Tables 8 and 9). We further performed the temperature-dependent conductivity measurements, as shown in Fig. 4g. Even at 30 °C, 100% RH, the proton conductivity of the resultant TpBD-(SO$_3$H)$_2$ iCOFMs reached 0.24 S cm$^{-1}$, on par with the proton conductivity of Nafion 117 at 80 °C, 100% RH[3]. Furthermore, we fabricated previously reported

iCOFMs (TpPa-SO$_3$H, NUS-9, IEC: 3.2 mmol g$^{-1}$) from aldehyde monomer Tp and ionic amine monomer 2,5-diaminobenzenesulfonic acid (Pa-SO$_3$H) using the same method. Dual-activation condition could afford high crystalline TpPa-SO$_3$H iCOFMs within 24 h (Supplementary Figs. 19–26). The proton conductivity of the resultant TpPa-SO$_3$H iCOFMs reached 0.2 S cm$^{-1}$ at 90 °C, 100% RH (Supplementary Fig. 26), lower than that of the TpBD-(SO$_3$H)$_2$ iCOFMs. The superior proton conductivity performance of the TpBD-(SO$_3$H)$_2$ iCOFMs can be explained as follows: Unlike amorphous polyelectrolyte membrane, the rigid framework structure with high density of –SO$_3$H groups endowed the TpBD-(SO$_3$H)$_2$ iCOFMs with superhigh IEC of 4.6 mmol g$^{-1}$, Fig. 4h), while maintaining a low swelling ratio (Supplementary Table 7). The ultra-high density of monodispersed sulfonic acid groups along the one-dimensional proton-conducting channels within the membrane will lead to a shorter proton hop distance based on surface transport mechanism[45,46]. Moreover, owing to the higher IEC, the TpBD-(SO$_3$H)$_2$ iCOFMs could absorb more water molecules (Supplementary Figs. 7 and 25), which play a critical role in proton transport based on Grotthuss and vehicle mechanisms[47]. Accordingly, the TpBD-(SO$_3$H)$_2$ iCOFMs afforded low energy barrier paths ($E_a = 0.15$ eV, Fig. 4i) for proton transport.

Moreover, we have carried out the long-term operation test of membrane proton conductivity at 90 °C and 100% RH (Supplementary Fig. 27). During the test, the proton conductivity of the TpBD-(SO$_3$H)$_2$ iCOFMs only slightly decreased by about 6% to 0.62 S cm$^{-1}$ within the first 5 h, which was probably due to the instability of the amorphous regions of the membrane under high temperature and high humidity. The proton conductivity of TpBD-(SO$_3$H)$_2$ iCOFMs remained stable around 0.62 S cm$^{-1}$ within two weeks, verifying the membrane's excellent long-term thermal durability for further fuel cell applications.

Due to the requirement of higher temperature for high catalytic activity and minimal catalyst poisoning, a high proton conductivity at low RH is more desired from a practical standpoint. Humidity-dependent proton conduction test was conducted for the iCOFMs fabricated in this work and Nafion 212 membrane (Fig. 5a). The proton conductivity of both types of iCOFMs show less decrease from 100% RH to 40% RH compared with Nafion 212 membrane (Fig. 5b, $\Delta_1 > \Delta_2 > \Delta_3$). Specifically, the TpBD-(SO$_3$H)$_2$ iCOFMs exhibited a proton conductivity of 0.1 S cm$^{-1}$ under 40% RH at 40 °C, which was 10 times higher than that of the TpPa-SO$_3$H iCOFMs and 500 times higher than that of

Nafion 212 membrane. Based on the Grotthuss mechanism and the vehicle mechanism, existing models indicate that water in membranes plays a significant role in proton transport[3,45,48]. Attributed to the condensation effect of rigid nanochannels and high density, monodispersed sulfonic acid groups, the nanochannels within iCOFMs can efficiently retain the water molecules under low RH[3]. In contrast, due to the lack of water molecules, the flexible ion transport channel in Nafion 212 may become discontinuous under low RH conditions[47]. Moreover, the TpBD-(SO$_3$H)$_2$ iCOFMs possess six sulfonic acid groups in each frame unit, two times more than the TpPa-SO$_3$H iCOFMs (Fig. 5b). Accordingly, the TpBD-(SO$_3$H)$_2$ iCOFMs possess higher water retention capacity, leading to more continuous water channels under low RH, which can effectively facilitate the proton conduction. This high proton conductivity at low RH indicates the great promise of our De Novo designed TpBD-(SO$_3$H)$_2$ iCOFMs for practical applications.

## Discussion

In this study, we developed a dual-activation interfacial polymerization strategy to fabricate iCOFMs, achieving superhigh ion exchange capacity of 4.6 mmol g$^{-1}$. Through simultaneously activating the aldehyde monomers in the organic phase and the ionic amine monomers in water phase during the interfacial polymerization, the Schiff-base reaction at the water–organic interface was significantly accelerated. Fukui function was employed as the descriptor to analyze the monomer reactivity before and after activation. We found that the Fukui function for nucleophilic attack sites of aldehyde monomer was increased from 0.066 to 0.154, while the Fukui function for electrophilic attack sites of amine monomers was increased from 0.009 to 0.072, leading to the formation of robust and high crystalline TpBD-(SO$_3$H)$_2$ iCOFMs. Furthermore, owing to the superhigh IEC and highly contiuous ionic channels, our De Novo designed TpBD-(SO$_3$H)$_2$ iCOFMs exhibited superior proton conductivity of 0.66 S cm$^{-1}$ (90 °C, 100% relative humidity). Moreover, the TpBD-(SO$_3$H)$_2$ iCOFMs also showed excellent proton conductivity around 0.62 S cm$^{-1}$ during the 2-week long-term proton conductivity test (90 °C, 100% relative humidity). We anticipate that our membranes can find diverse applications, such as fuel cell, flow cell, lithium battery, and nanofiltration. Moreover, our dual-activation interface polymerization strategy can also inspire researchers in the field of chemical bond activation,

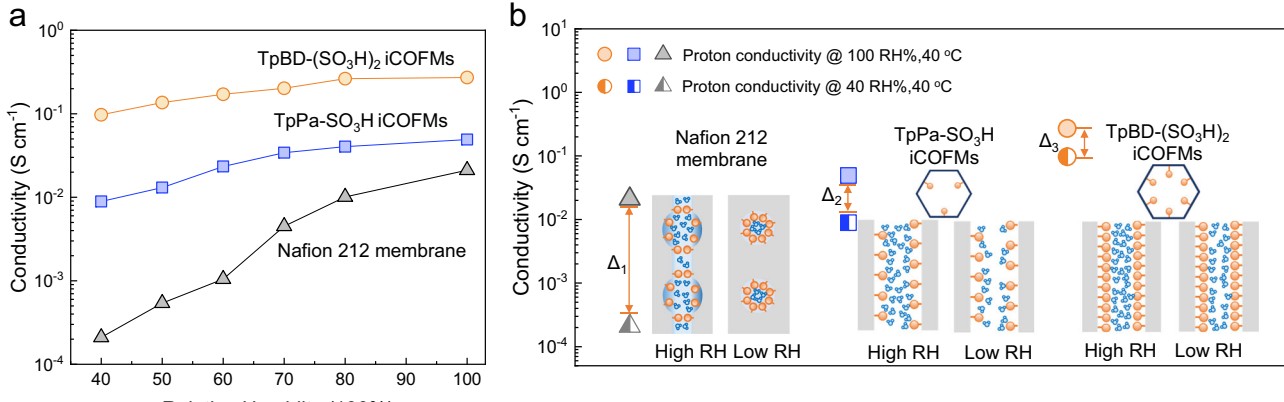

**Fig. 5 Humidity–conductivity relationship of TpPa-SO$_3$H, TpBD-(SO$_3$H)$_2$ iCOFMs, and Nafion 212 membrane. a** Proton conductivity of the TpPa-SO$_3$H, TpBD-(SO$_3$H)$_2$ iCOFMs, and Nafion 212 membrane versus relative humidity at 40 °C. **b** Schematic illustration of ion transport channels changes with decreased relative humidity (RH) for the Nafion 212 membrane, TpPa-SO$_3$H and TpBD-(SO$_3$H)$_2$ iCOFMs, $\Delta$ represents the difference in the proton conductivity of the membranes at 100% RH and 40% RH.

and is expected to become a platform technology for organic framework membrane fabrication.

## Methods

**Materials and chemicals**. 2,4,6-triformylphloroglucinol (Tp) was bought form Jilin Chinese Academy of Sciences—Yanshen Technology Co., Ltd., China. 4,4′-diamino-biphenyl-3,3′-disulfonic acid (BD-(SO$_3$H)$_2$) was bought from Zhengzhou Alfachem Inc., China. 2,5-diaminobenzenesulfonic acid (Pa-SO$_3$H), n-octanoic acid, n-heptanoic acid, n-nonanoic acid, and n-decanoic acid, sodium formate, sodium chloride sodium acetate and sodium benzoate were bought from TCI and aladdin chemicals. All commercially available chemicals and solvents were used without further purification.

**Fabrication of the TpBD-(SO$_3$H)$_2$ iCOFMs**. 0.2 mmol of 2,4,6-triformylphlor-oglucinol (Tp) was dissolved into a mixed solvent of n-octanoic acid and mesi-tylene (x mL n-octanoic acid/20−x mL mesitylene), in which n-octanoic acid was the aldehyde monomer activator, and mesitylene has no activating function for aldehyde monomers. 0.3 mmol (103.3 mg) of 4,4′-diaminobiphenyl-3,3′-dis-ulphonic acid (BD-(SO$_3$H)$_2$) and 0–1.05 mmol (0–3.5 equivalent) of sodium for-mate was dissolved into 30 mL deionized water by sonication for 20 min. The amine monomer solution was poured into the bottom of 100 mL beaker, and the Tp solution was added by drops on the top layer. Under static conditions for 24 h, the TpBD-(SO$_3$H)$_2$ iCOFMs could be taken out from the beaker using tweezers. The diameter of iCOFMs was 4.7 cm, same as the inner diameter of the 100 mL beaker. After washing with DMF and ethyl alcohol, the TpBD-(SO$_3$H)$_2$ iCOFMs was taken out for further characterization and performances evaluation.

**Synthesis of the TpBD-(SO$_3$H)$_2$ powder**. Typically, 0.2 mmol (42.0 mg) of 2,4,6-triformylphloroglucinol (Tp), 0.3 mmol (103.3 mg) of 4,4′-diaminobiphenyl-3,3′-disulfonic acid (BD-(SO$_3$H)$_2$), 3 mL mesitylene, 1 mL dioxane and acetic acid (6 mol L$^{-1}$, 0.5 mL) were added in a 15 mL pyrex tube. The mixture was sonicated for 20 min and frozen in liquid nitrogen, followed by degassing treatment by three freeze–pump–thaw cycles. Then, the tube was sealed and heated at 120 °C for 3 days. The deep red powder was collected by filtration, washed with anhydrous ethanol, anhydrous acetone and deionized water for three times and subjected to Soxhlet extraction with methanol for 3 days. Finally, the deep red powder was collected after drying at 120 °C under vacuum. Yield: 107.8 mg, 74.2%.

**Characterizations**. The powder X-ray diffraction (PXRD) data were collected on a Rigaku D/max 2500 v/pc diffractometer. The Fourier transform infrared (FT-IR) spectra were recorded on a BRUKER Vertex 70 spectrometer. Scanning electron microscopy (SEM) images were collected by field emission scanning electron microscope (Nanosem 430) and transmission electron microscopy (TEM) images were obtained by HRTEM (Tecnai G2 F20). Solid-state $^{13}$C NMR spectra were recorded on Bruker 600 MHz NMR spectrometer (JEOL JNM ECZ600R). N$_2$ sorption isotherms were recorded on PS2-1055-B gas adsorption analyzers at 77 K using a liquid nitrogen bath, vacuum water vapor sorption isotherm was recorded on 3H-2000PW gas adsorption analyzers. GIXRD measurement was performed by Rigaku Smartlab 9KW diffractometer with 0.5° incident angle. GIWAXS mea-surement was performed by a Mars345 detector with X-ray wavelength of 0.15418 nm, the incident angle was 0.5° and the distance of sample to ditector was 438 mm. AFM (Dimension Icon, Bruker Co. Ltd., USA) analysis was utilized to characterize the surface roughness of the membrane. The stress–strain curves of the membranes were obtained by a Yangzhou Zhongke WDW-02 electronic stretching machine with a strain rate of 2 mm min$^{-1}$ under ambient condition. Thermo-gravimetric analysis (TGA) was carried out on a NETZSCH 209F3 thermal analyzer at a heating rate of 10 °C min$^{-1}$ min from 40 to 800 °C and under N$_2$ atmosphere.

**Proton conductivity measurement**. In-plane proton conductivity of the iCOFMs was measured by a two-probe method. The alternating current impedance measure-ments were performed using an impedance/gain-phase analyzer (PARSTAT4000) in the 1 MHz to 10 Hz range with a signal amplitude of 15 mV. The measurements were conducted at temperatures between 30 and 90 °C inside a thermos-hygrostat. The proton conductivities ($\sigma$, S cm$^{-1}$) were calculated as follows:

$$\sigma = \frac{l}{AR} \quad (1)$$

were $l$ (cm), $A$ (cm$^2$) and $R$ ($\Omega$) is the distance between the electrodes, the cross-section area and Ohmic resistance of the membranes, respectively. The cross-sectional area $A$ of membranes was calculated by the membrane width times the membrane thickness.

**Monomer reactivity calculation**. In this study, Fukui function was calculated for monomers reactivity discussed. The Fukui function were calculated based on Hirshfeld population analysis as follows:

$$\text{Electrophilic reaction}: f_A^- = q_{N+1}^A - q_N^A \quad (2)$$

$$\text{Nucleophilic reaction}: f_A^+ = q_N^A - q_{N-1}^A \quad (3)$$

where $q_N^A$ is the charge of atom $A$ in a molecule. In this work, geometry optimi-zations, Mulliken charges calculation and generation of electronic wavefunctions were done with Gaussian 16 program at level of B3LYP/6-31G*[49]. Fukui function were calculations including image drawing were carried out with Multiwfn[50]. Fukui function of monomers was shown in Supplementary Tables 2–6. In general, the larger the value of the Fukui function at a reaction site, the higher the reactivity of that corresponding site[49]. It should be noted that some of the calculated results of chemical equivalent reaction sites are not equivalent. In this case, an average of results was taken and used for all equivalent reaction sites.

## Data availability

All data supporting the findings of this study are available within the article and the Supplementary Information file, or available from the corresponding authors upon request. The crystallographic data of TpBD-(SO$_3$H)$_2$ COF have been deposited in The Cambridge Crystallographic Data Centre under the reference number CCDC 2089523. This data can be obtained free of charge from The Cambridge Crystallographic Data Centre via www.ccdc.cam.ac.uk/data_request/cif. Source data are provided with this paper.

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

## Acknowledgements

The authors gratefully acknowledge financial support from National Natural Science Foundation of China, grant No. 21838008 (Z.J.), 22008172 (R.Z.), U20B2024 (H.W.), 22103054 (H.Y.) and 21903058 (T.C.). R.Z. acknowledges the support by China Post-doctoral Science Foundation (Grant Nos. 2020TQ0226 and 2021M692384). T.C. acknowledges the support of Natural Science Foundation of Jiangsu Higher Education Institutions (Grant No. BK20190810), and Jiangsu Province High-Level Talents (Grant No. JNHB-106). We thank Bohui Lv for the help on structure simulation of COFs.

## Author contributions

X.W. and B.S. conceived the idea and designed the experiments. J.G., C.F., X.Y., Y.W., H.W., R.Z. and Z.J. provided guidance on the experimental design and article writing. H.Y., Z.Z., T.C. carried out the Fukui function calculation and analyzed the data. X.W., B.S. and X.L. performed the fabrication and characterization ionic COF membranes. All authors contributed to the discussion and analysis of the results.

## Competing interests

The authors declare no competing interests.
