## [Peer Review File · Nature Communications]

Assembling covalent organic framework membranes with superhigh ion exchange capacityREVIEWER COMMENTS

Reviewer #1 (Remarks to the Author):

It is an interesting article which should be published after minor revision.

1. Correct in Fig. 1g cm³
2. Fukui function is not so common. Say in the main text one sentence what is the Fukui function.
3. There is some inconsistency in the use of bolds in the figures. See the ordinate of Fig. 2e and f: "Fukui functions of Nucleo/Electrophilic Attack". Why function not bold?
4. Authors addressed the high mechanical strength of 20 MPa. Membranes can have different thickness between 3 and almost 100 μm . Does it mean that membranes can be used without support? Which thickness can be used without support?
5. Conclusions tell the same information like Abstract, however with different words. Can the authors speculate in Conclusions more about possible applications, scale up, costs, regeneration, life cycle assessment...
6. Literature citations need revision.
 - In ref. 33,34,35 the journals are with full name, in 36,37,38 the names are abbreviated.
 - Example: 35 is Nature Communications, in 43 Nat Commun
 - Example: Angew. Chem. Int. Ed. (this is the correct name) is found in 4 different writings (Angew Chem Int Ed Engl, Angewandte Chemie International Edition, Angewandte Chemie International Edition n/a, ...

Reviewer #2 (Remarks to the Author):

In this manuscript, the authors reported the dual-activation interfacial polymerization strategy for the preparation of ionic covalent organic framework membranes (iCOFMs) with the highest ion exchange capacity ever reported. The iCOFMs exhibited higher crystallinity compared with the same type of COF powder. In addition, the membrane showed the highest conductivity, which was superior to other reported COF materials. However, some results are still missing in the manuscript to fully support the conclusions. I recommend to reconsider the manuscript after authors addressing the following points:

1. The defect-free iCOFMs exhibited an asymmetrical structure. The iCOFMs toward oil phase side showed a smooth and defect-free membrane surface while the iCOFMs toward water phase side showed loose membrane structure. Please discuss what causes this difference.
2. The experimental PXRD pattern was highly consistent with the simulated reversed AA stacking PXRD patterns with the existence of the sharp diffraction peaks corresponding to the 100 planes. Where are the sharp peaks located, and which crystal planes do the other peaks correspond to?
3. The HRTEM images of the slices of iCOFMs verified the ordered crystalline structures. The pore size of TpBD-(SO₃H)₂ should be larger than that of TpPa-SO₃H whose pore size is 1.4 nm (Adv. Mater. 2020, 2005565; ACS Appl. Mater. Interfaces 2016, 8, 18505–18512). Are the results in Fig. 1f consistent with the crystal structure of TpBD-(SO₃H)₂?
4. For most state-of-the-art PEMs, increasing IEC value will impart higher water uptake, which in turn causes higher swelling ratio, originating from the flexible polymer networks nature (Adv. Mater. 2020, 2005565). But the rigid structure of iCOFMs is likely to reduce the swelling ratio. It is recommended to provide water uptake and swelling properties of the prepared membranes.
5. Why do two sharp peaks appear in Fig 3b, e? Please clarify.
6. The iCOFMs had the high structural stability. In order to investigate the stability of the material under high humidity and high temperature conditions, it is necessary to run the membrane at 90 °C and 100% RH for a continuous period of time to prove the long-term thermal durability for further fuel cell applications.

7. Please unify whether the abbreviation for 2,4,6-triformylphloroglucinol is TP or Tp, and check the written errors in the manuscript.

Reviewer #3 (Remarks to the Author):

This work reports the synthesis of iCOF membranes by a dual-activation strategy and demonstrates high proton conductivity of membranes. The COF membrane formation mechanism is profoundly studied and discussed. However, the ion exchange capacity tests and discussions are not profound enough. The reported results are also similar to their previous works despite the high proton conductivity. Overall, the work may not satisfy the high quality of Nature Communication. The detailed comments are listed below.

1. The increase of n-octanoic acid amount results in raised membrane thicknesses, leading to decreased proton conductivity. From Figure 3b, it can be seen that the membrane crystallinity is strongly influenced by the amount of n-octanoic acid. Thus, the crystallinity of these membranes may affect the proton conductivity as well.
2. The synthesized TpPa-SO₃H membranes show lower proton conductivity than TpBD-(SO₃H)₂. Please explain this result.
3. The dual-activation polymerization is illustrated in this manuscript. Sodium formate and n-octanoic acid are selected as the activator pairs. The selection criteria should be discussed to inspire others. Are there any other viable activators?
4. The weight loss in 40-150°C is ~85%. This is very considerable, which may not arise from the evaporation of absorbed water.
5. It is interesting to find that thus-prepared iCOF membranes hold higher crystallinity than their bulk powders synthesized under solvothermal conditions. It will be valuable to guide the synthesis of highly crystalline COF membranes if the underlying mechanism is discussed.
6. Given the asymmetric structure of membranes, i.e. top dense films and loosely stacked particles, the membrane may contain a crystallinity asymmetry. The crystallinity of the film and particles should be presented separately.
7. The sulfonic acid groups can be easily introduced into COF skeletons by post-synthetic strategies. The advantage of this De Novo method is suggested to be added.
8. The HRTEM image in Figure 1f is not clear, please enhance the resolution.
9. There are some grammatical errors that need revision.
10. The iCOF membranes give higher proton conductivities than others and commercial materials. How about the stability of proton conductivity?

Reviewer #1 (Remarks to the Author):

It is an interesting article which should be published after minor revision.

Reply:

Thanks the reviewer for the highly positive remarks.

1. Correct in Fig. 1g cm³

Reply:

Following this reviewer's valuable guidance, we have corrected "cm₃" as "cm³" in Fig. 1g of the revised manuscript, as shown below.

Figure 1. (g) N₂ sorption isotherms of iCOFMs

2. Fukui function is not so common. Say in the main text one sentence what is the Fukui function.

Reply:

Thanks for the reviewer's valuable guidance. We have added one sentence about Fukui Function in the Introduction section. "...Fukui function, first proposed by Parr and Yang in 1984, has been widely applied as a tool for deducing the relative reactivity of different positions in a molecule⁴¹. In this regard, we employed the Fukui function as a descriptor to compare the reactivity of amine monomers with different ionic groups..."

3. There is some inconsistency in the use of bolds in the figures. See the ordinate of Fig. 2e and f: "Fukui functions of Nucleo/Electrophilic Attack". Why function not bold?

Reply:

Thanks for the reviewer's valuable guidance. We have carefully checked the text in Figure 2e-f and unified the font in the revised manuscript, as shown below.

Figure 2. (e) Fukui functions for nucleophilic attack sites (f^+) of aldehyde monomer; (f) Fukui functions for electrophilic attack sites (f^-) of amine monomer.

4. Authors addressed the high mechanical strength of 20 MPa. Membranes can have different thickness between 3 and almost 100 μm . Does it mean that membranes can be used without support? Which thickness can be used without support?

Reply:

Thanks for the reviewer's valuable guidance. All iCOFMs in our manuscript were used and tested without support. Figure S16 is the digital photograph of the TpBD-(SO₃H)₂ membranes. When the membrane thickness exceeds 3 μm , the mechanical strength of the membrane is strong enough for testing without support. Only when the membrane thickness is as low as 0.5 μm , the mechanical strength of the membrane cannot be tested without support.

Figure S16. Effect of the addition amount of amine monomer activator on the thickness of TpBD-(SO₃H)₂ membranes.

5. Conclusions tell the same information like Abstract, however with different words. Can the authors speculate in Conclusions more about possible applications, scale up, costs, regeneration, life cycle assessment...

Reply:

Thanks for the reviewer's valuable guidance. We have added some speculations about possible applications in the Conclusions section of the revised manuscript. "...outperforming all the currently reported COF-based proton-conducting materials. We anticipate that our membranes can find

diverse applications such as fuel cell, flow cell, lithium battery and nanofiltration. Moreover, our dual activation interface polymerization strategy can also inspire researchers in the field of chemical bond activation, and is expected to become a platform technology for organic framework membrane fabrication.”

6. Literature citations need revision.

- In ref. 33,34,35 the journals are with full name, in 36,37,38 the names are abbreviated.

- Example: 35 is Nature Communications, in 43 Nat Commun

- Example: Angew. Chem. Int. Ed. (this is the correct name) is found in 4 different writings (Angew Chem Int Ed Engl, Angewandte Chemie International Edition, Angewandte Chemie International Edition n/a, ...

Reply:

Thanks for the reviewer’s valuable guidance. We have carefully checked and corrected the journal names, as shown below.

- 1 Huang, N., Wang, P. & Jiang, D. Covalent organic frameworks: a materials platform for structural and functional designs. *Nat. Rev. Mater.* **1**, 16068, doi:10.1038/natrevmats.2016.68 (2016).
- 2 Chen, X. *et al.* Covalent organic frameworks: chemical approaches to designer structures and built-in functions. *Angew. Chem. Int. Edit.* **59**, 5050, doi:10.1002/anie.201904291 (2019).
- 3 Cao, L. *et al.* Weakly humidity-dependent proton-conducting COF membranes. *Adv. Mater.* **32**, 2005565, doi:10.1002/adma.202005565 (2020).
- 4 He, X. *et al.* De Novo Design of Covalent Organic Framework Membranes toward Ultrafast Anion Transport. *Adv. Mater.* **32**, 2001284, doi:10.1002/adma.202001284 (2020).
- 5 Xiong, X. H. *et al.* Selective extraction of thorium from uranium and rare earth elements using sulfonated covalent organic framework and its membrane derivate. *Chem. Eng. J.* **384**, 123240, doi:10.1016/j.cej.2019.123240 (2020).
- 6 Xiong, X. H. *et al.* Ammoniating Covalent Organic Framework (COF) for High-Performance and Selective Extraction of Toxic and Radioactive Uranium Ions. *Adv. Sci.* **6**, 1900547, doi:10.1002/advs.201900547 (2019).
- 7 Sun, Q. *et al.* Covalent Organic Frameworks as a Decorating Platform for Utilization and Affinity Enhancement of Chelating Sites for Radionuclide Sequestration. *Adv. Mater.* **30**, 1705479, doi:10.1002/adma.201705479 (2018).
- 8 Chen, H. *et al.* Cationic Covalent Organic Framework Nanosheets for Fast Li-Ion Conduction. *J. Am. Chem. Soc.* **140**, 896, doi:10.1021/jacs.7b12292 (2018).
- 9 Huang, N., Chen, X., Krishna, R. & Jiang, D. Two-dimensional covalent organic frameworks for carbon dioxide capture through channel-wall functionalization. *Angew. Chem. Int. Edit.* **54**, 2986, doi:10.1002/anie.201411262 (2015).
- 10 Zhang, P., Wang, Z., Cheng, P., Chen, Y. & Zhang, Z. Design and application of ionic covalent organic frameworks. *Coord. Chem. Rev.* **438**, 213873, doi:10.1016/j.ccr.2021.213873 (2021).
- 11 Wang, H. *et al.* Organic molecular sieve membranes for chemical separations. *Chem. Soc. Rev.* **50**, 5468, doi:10.1039/D0CS01347A (2021).
- 12 Jeong, K. *et al.* Solvent-Free, Single Lithium-Ion Conducting Covalent Organic Frameworks. *J. Am. Chem. Soc.* **141**, 5880, doi:10.1021/jacs.9b00543 (2019).
- 13 Hou, S. *et al.* Free-Standing Covalent Organic Framework Membrane for High-Efficiency Salinity Gradient Energy Conversion. *Angew. Chem. Int. Edit.* **60**, 9925, doi:10.1002/anie.202100205 (2021).
- 14 Liu, L. *et al.* Surface-Mediated Construction of Ultrathin Free-standing Covalent Organic Framework Membrane for Efficient Proton Conduction. *Angew. Chem. Int. Edit.* **60**, 14875, doi:10.1002/anie.202104106 (2021).
- 15 Chandra, S. *et al.* Interplaying Intrinsic and Extrinsic Proton Conductivities in Covalent Organic Frameworks. *Chem. Mat.* **28**, 1489, doi:10.1021/acs.chemmater.5b04947 (2016).

- 16 Hou, L. *et al.* Understanding the Ion Transport Behavior across Nanofluidic Membranes in Response to
the Charge Variations. *Adv. Funct. Mater.* **31**, 2009970, doi:10.1002/adfm.202009970 (2021).
- 17 Zhang, W., Zhang, L., Zhao, H., Li, B. & Ma, H. A two-dimensional cationic covalent organic framework
membrane for selective molecular sieving. *J. Mater. Chem. A* **6**, 13331, doi:10.1039/c8ta04178d (2018).
- 18 Kong, Y. *et al.* Tight covalent organic framework membranes for efficient anion transport via molecular
precursor engineering. *Angew. Chem. Int. Edit.* **60**, 17638, doi:10.1002/anie.202105190 (2021).
- 19 Dey, K. *et al.* Selective Molecular Separation by Interfacially Crystallized Covalent Organic Framework
Thin Films. *J. Am. Chem. Soc.* **139**, 13083, doi:10.1021/jacs.7b06640 (2017).
- 20 Kandambeth, S. *et al.* Selective Molecular Sieving in Self-Standing Porous Covalent-Organic-Framework
Membranes. *Adv. Mater.* **29**, 1603945, doi:10.1002/adma.201603945 (2017).
- 21 Sasmal, H. S. *et al.* Superprotonic Conductivity in Flexible Porous Covalent Organic Framework
Membranes. *Angew. Chem. Int. Edit.* **57**, 10894, doi:10.1002/anie.201804753 (2018).
- 22 Kandambeth, S., Dey, K. & Banerjee, R. Covalent Organic Frameworks: Chemistry beyond the Structure.
J. Am. Chem. Soc. **141**, 1807, doi:10.1021/jacs.8b10334 (2019).
- 23 Dey, K., Bhunia, S., Sasmal, H. S., Reddy, C. M. & Banerjee, R. Self-Assembly-Driven Nanomechanics in
Porous Covalent Organic Framework Thin Films. *J. Am. Chem. Soc.* **143**, 955, doi:10.1021/jacs.0c11122
(2021).
- 24 Matsumoto, M. *et al.* Lewis-Acid-Catalyzed Interfacial Polymerization of Covalent Organic Framework
Films. *Chem* **4**, 308, doi:10.1016/j.chempr.2017.12.011 (2018).
- 25 Fan, C. *et al.* Scalable fabrication of crystalline COF membrane from amorphous polymeric membrane.
Angew. Chem. Int. Edit. **60**, 18051, doi:10.1002/anie.202102965 (2021).
- 26 Fenton, J. L., Burke, D. W., Qian, D., Cruz, M. O. & Dichtel, W. R. Polycrystalline Covalent Organic
Framework Films Act as Adsorbents, Not Membranes. *J. Am. Chem. Soc.* **143**, 1466,
doi:10.1021/jacs.0c11159 (2021).
- 27 Burke, D. W. *et al.* Acid Exfoliation of Imine-linked Covalent Organic Frameworks Enables Solution
Processing into Crystalline Thin Films. *Angew. Chem. Int. Edit.* **59**, 5165, doi:10.1002/anie.201913975
(2020).
- 28 Wang, H. *et al.* Recent progress in covalent organic framework thin films: fabrications, applications and
perspectives. *Chem. Soc. Rev.* **48**, 488, doi:10.1039/c8cs00376a (2018).
- 29 Yuan, S. *et al.* Covalent organic frameworks for membrane separation. *Chem. Soc. Rev.* **48**, 2665,
doi:10.1039/c8cs00919h (2019).
- 30 Chen, S. *et al.* Imparting Ion Selectivity to Covalent Organic Framework Membranes Using de Novo
Assembly for Blue Energy Harvesting. *J. Am. Chem. Soc.* **143**, 9415, doi:10.1021/jacs.1c02090 (2021).
- 31 Liu, J. *et al.* Self-standing and flexible covalent organic framework (COF) membranes for molecular
separation. *Sci. Adv.* **6**, eabb1110, doi:10.1126/sciadv.abb1110 (2020).
- 32 Shao, P. *et al.* Flexible Films of Covalent Organic Frameworks with Ultralow Dielectric Constants under
High Humidity. *Angew. Chem. Int. Edit.* **57**, 16501, doi:10.1002/anie.201811250 (2018).
- 33 Shen, J. *et al.* Polydopamine-modulated covalent organic framework membranes for molecular
separation. *J. Mater. Chem. A* **7**, 18063, doi:10.1039/c9ta05040j (2019).
- 34 Yin, C., Fang, S., Shi, X., Zhang, Z. & Wang, Y. Pressure-modulated synthesis of self-repairing covalent
organic frameworks (COFs) for high-flux nanofiltration. *J. Membr. Sci.* **618**, 118727,
doi:10.1016/j.memsci.2020.118727 (2021).
- 35 Yang, H. *et al.* Covalent organic framework membranes through a mixed-dimensional assembly for
molecular separations. *Nat. Commun.* **10**, 2101, doi:10.1038/s41467-019-10157-5 (2019).
- 36 Banerjee, R., Dey, K., Kunjattu, H. S. & A, M. C. Nanoparticle Size-Fractionation through Self-Standing
Porous Covalent Organic Framework Films. *Angew. Chem. Int. Edit.* **59**, 1161,
doi:10.1002/anie.201912381 (2019).
- 37 Kandambeth, S. *et al.* Construction of crystalline 2D covalent organic frameworks with remarkable
chemical (acid/base) stability via a combined reversible and irreversible route. *J. Am. Chem. Soc.* **134**,
19524, doi:10.1021/ja308278w (2012).
- 38 Peng, Y. *et al.* Mechanoassisted Synthesis of Sulfonated Covalent Organic Frameworks with High Intrinsic
Proton Conductivity. *ACS Appl. Mater. Interfaces* **8**, 18505, doi:10.1021/acsami.6b06189 (2016).
- 39 Chen, T. *et al.* Highly crystalline ionic covalent organic framework membrane for nanofiltration and
charge-controlled organic pollutants removal. *Sep. Purif. Technol.* **256**, 117787,
doi:10.1016/j.seppur.2020.117787 (2021).
- 40 Karak, S., Kumar, S., Pachfule, P. & Banerjee, R. Porosity Prediction through Hydrogen Bonding in
Covalent Organic Frameworks. *J. Am. Chem. Soc.* **140**, 5138, doi:10.1021/jacs.7b13558 (2018).
- 41 Parr, R. G. & Yang, W. Density functional approach to the frontier-electron theory of chemical reactivity.
J. Am. Chem. Soc. **106**, 4049, doi:10.1021/ja00326a036 (1984).
- 42 Smith, M. B. & March, J. *MARCH'S ADVANCED ORGANIC CHEMISTRY*. Sixth Edition edn, 1251-1253 (John
Wiley & Sons, 2007).
- 43 Biswal, B. P. *et al.* Mechanochemical synthesis of chemically stable isoreticular covalent organic
frameworks. *J. Am. Chem. Soc.* **135**, 5328, doi:10.1021/ja4017842 (2013).

- 44 Li, Y. *et al.* Laminated self-standing covalent organic framework membrane with uniformly distributed subnanopores for ionic and molecular sieving. *Nat. Commun.* **11**, 599, doi:10.1038/s41467-019-14056-7 (2020).
- 45 Peckham, T. J. & Holdcroft, S. Structure-Morphology-Property Relationships of Non-Perfluorinated Proton-Conducting Membranes. *Adv. Mater.* **22**, 4667, doi:10.1002/adma.201001164 (2010).
- 46 Eikerling, M. & Kornyshev, A. A. Proton transfer in a single pore of a polymer electrolyte membrane. *Journal of Electroanalytical Chemistry* **502**, 1, doi:10.1016/S0022-0728(00)00368-5 (2001).
- 47 Hickner, M. A., Ghassemi, H., Kim, Y. S., Einsla, B. R. & McGrath, J. E. Alternative Polymer Systems for Proton Exchange Membranes (PEMs). *Chem. Rev.* **104**, 4587, doi:10.1021/cr020711a (2004).
- 48 Kreuer, K.-D., Paddison, S. J., Spohr, E. & Schuster, M. Transport in Proton Conductors for Fuel-Cell Applications: Simulations, Elementary Reactions, and Phenomenology. *Chem. Rev.* **104**, 4637, doi:10.1021/cr020715f (2004).
- 49 Cao, J., Ren, Q., Chen, F. & Lu, T. Comparative study on the methods for predicting the reactive site of nucleophilic reaction. *Sci. China Chem.* **58**, 1845, doi:10.1007/s11426-015-5494-7 (2015).
- 50 Lu, T. & Chen, F. Multiwfn: a multifunctional wavefunction analyzer. *J. Comput. Chem.* **33**, 580, doi:10.1002/jcc.22885 (2012).

Reviewer #2 (Remarks to the Author):

In this manuscript, the authors reported the dual-activation interfacial polymerization strategy for the preparation of ionic covalent organic framework membranes (iCOFMs) with the highest ion exchange capacity ever reported. The iCOFMs exhibited higher crystallinity compared with the same type of COF powder. In addition, the membrane showed the highest conductivity, which was superior to other reported COF materials. However, some results are still missing in the manuscript to fully support the conclusions. I recommend to reconsider the manuscript after authors addressing the following points:

1. The defect-free iCOFMs exhibited an asymmetrical structure. The iCOFMs toward organic phase side showed a smooth and defect-free membrane surface while the iCOFMs toward water phase side showed loose membrane structure. Please discuss what causes this difference.

Reply:

Thanks for the reviewer's valuable guidance. When the interfacial polymerization started, the activated amine monomers diffused into the organic phase and reacted with the activated aldehyde monomers at the water-organic interface. After dual-activation, the higher reaction rate could generate COF particles instead of continuous COF membranes (*Sci. Adv.* 2020, 6, eabb1110). As the reaction time prolonged, the COF particles will stack into a loose layer, which reduced the further diffusion rate of amine monomers into the organic phase. This is similar to the self-sealing and self-terminating behavior in the interfacial polymerization process for polyamide membranes (*J. Mater. Chem. A*, 2019, 7, 25641; *J. Mater. Chem. A*, 2020, 8, 23930). Accordingly, the interfacial polymerization was slowed down, leading to the formation of a compacted layer on the top side. Therefore, the resultant iCOFMs exhibited an asymmetrical structure with a smooth and defect-free membrane surface toward organic phase side and a loose membrane structure toward water phase side.

The relevant discussion was added in the revised manuscript. "...When the interfacial polymerization started, the activated BD-(SO₃H)₂ monomers diffused into the organic phase and reacted with the activated Tp monomers at the water-organic interface. After dual-activation, the higher reaction rate could generate COF particles instead of continuous COF membranes³¹. As the reaction time prolonged, the COF particles will stack into a loose layer, which reduced the further diffusion rate of amine monomers into the organic phase. Accordingly, the interfacial polymerization was slowed down, leading to the formation of a compacted layer on top side. After 24 hours, defect-free iCOFMs finally formed, which exhibited an asymmetrical structure (Figure S1, Supplementary Information). The SEM and AFM images of the iCOFMs toward organic phase side showed a smooth and defect-free membrane surface with a roughness of 8.20 nm (Figure 1b)..."

2. The experimental PXRD pattern was highly consistent with the simulated reversed AA stacking PXRD patterns with the existence of the sharp diffraction peaks corresponding to the 100 planes. Where are the sharp peaks located, and which crystal planes do the other peaks correspond to?

Reply:

Thanks for the reviewer's valuable guidance. The crystal plane corresponding to the diffraction peak has been inserted in Figure 2d in the revised manuscript, as show below. The diffraction peak at 3.3° , 5.7° and 26° were corresponding to 100, 110 and 002 crystal plane, respectively.

Figure 1. (d) Experimental PXRD pattern, and simulated PXRD patterns for AA and reversed AA stacking of iCOFMs.

3. The HRTEM images of the slices of iCOFMs verified the ordered crystalline structures. The pore size of TpBD-(SO₃H)₂ should be larger than that of TpPa-SO₃H whose pore size is 1.4 nm (Adv. Mater. 2020, 2005565; ACS Appl. Mater. Interfaces 2016, 8, 18505–18512). Are the results in Fig. 1f consistent with the crystal structure of TpBD-(SO₃H)₂?

Reply:

Thanks for the reviewer's valuable guidance. A HTEM image with higher resolution has been presented in Figure 1f in the revised manuscript. We can see that the 002 crystal plane width was 0.34 nm, consistent with the crystal plane obtained by simulation.

Figure 1. (f) High-resolution TEM images of the slices of iCOFMs

4. For most state-of-the-art PEMs, increasing IEC value will impart higher water uptake, which in turn causes higher swelling ratio, originating from the flexible polymer networks nature (Adv.

Mater. 2020, 2005565). But the rigid structure of iCOFMs is likely to reduce the swelling ratio. It is recommended to provide water uptake and swelling properties of the prepared membranes.

Reply:

Thanks for the reviewer's valuable guidance. We further tested the water uptake and swelling properties of the resulting iCOFMs (Figure S18, Table S7). The water uptake of TpBD-(SO₃H)₂ membrane was 144 ± 5%, higher than most state-of-the-art PEMs. This was due to the high IEC value as well as the high specific surface area of the TpBD-(SO₃H)₂ membrane. Moreover, the TpBD-(SO₃H)₂ membrane exhibited the area and thickness swelling ratio of 18.3 ± 2.9% and 20.0 ± 2.4%, respectively, lower than most state-of-the-art PEMs. This was because the rigid framework structure of the TpBD-(SO₃H)₂ membrane was likely to considerably restrain the swelling.

Figure S18. Digital photographs of wet and dry TpBD-(SO₃H)₂ membranes.

Table S7. Swelling ratio and water uptake of iCOFMs and membranes reported in literature.

Membranes	IEC (mmol g ⁻¹)	Swelling ratio (%)		Temperature (°C)	Conditions	Water uptake (%)	Ref.
		Thickness	Area				
TpBD-(SO ₃ H) ₂	4.6	22	21	25	DI water	144.4	This work
TpBD-(SO ₃ H) ₂		25	23	80			
TpPa-SO ₃ H	3.2	11	14	25		75.3	This work
TpPa-SO ₃ H		15	16	80			
IPC-COF membrane	3.2	9	1	25		60	¹
Random copolymer	3.28	110	33	80		183.3	²
	3.19	77	38	80		119	²
	3.26	160	132	50		44	³
	2.51	95	180	50		35	³
Block polymer	1.83	70	25	20		91.2	⁴
	1.50	65	11	30	48	⁵	
	1.67	55	16	25	82	⁶	
	1.83	10		20	25	⁷	
Nafion 212®	0.93	14	14	25	12	¹	
	0.93	21	44	80	36		

5. Why do two sharp peaks appear in Fig 3b, e? Please clarify.

Reply:

Thanks for the reviewer's valuable guidance. We re-tested the iCOFM samples involved in Figure 3b, e for three times, and found only one spike in the re-tested XRD pattern. The peak at 2.6 degrees in the previous test may be caused by sample contamination. The XRD patterns in Figure 3b, e were updated in the revised manuscript as shown below.

Figure 3. (b) Effect of the aldehyde monomer activator addition amount on crystallinity of iCOFMs; (e) Effect of the amine monomer activator addition amount on crystallinity of iCOFMs.

6. The iCOFMs had the high structural stability. In order to investigate the stability of the material under high humidity and high temperature conditions, it is necessary to run the membrane at 90 °C and 100% RH for a continuous period of time to prove the long-term thermal durability for further fuel cell applications.

Reply:

Following the reviewer's valuable guidance, we have carried out the long-term operation test of membrane proton conductivity at 90 °C and 100% RH. The results were added in Figure S27 in the revised manuscript, as shown below. During the test, and the proton conductivity of the TpBD-(SO₃H)₂ membrane only slightly decreased by about 7.6% within the first 5 hours, which was probably due to the instability of the amorphous regions of the membrane under high temperature and high humidity, and then remained stable within two weeks, verifying the membrane's excellent long-term thermal durability for further fuel cell applications.

The relevant discussion was added in the revised manuscript. "...Moreover, we have carried out the long-term operation test of membrane proton conductivity at 90 °C and 100% RH (Figure S27). During the test, and the proton conductivity of the TpBD-(SO₃H)₂ iCOFMs only slightly decreased by about 7.6% within the first 5 hours, which was probably due to the instability of the amorphous regions of the membrane under high temperature and high humidity, and then remained stable within two weeks, verifying the membrane's excellent long-term thermal durability for further fuel cell applications."

Figure S27. long-term proton conductivity of the TpBD-(SO₃H)₂ membrane

7. Please unify whether the abbreviation for 2,4,6-triformylphloroglucinol is TP or Tp, and check the written errors in the manuscript.

Reply:

Thanks for the reviewer's valuable guidance. We have checked the written errors and unified the abbreviation for 2,4,6-triformylphloroglucinol as Tp in the revised manuscript.

Reviewer #3 (Remarks to the Author):

This work reports the synthesis of iCOF membranes by a dual-activation strategy and demonstrates high proton conductivity of membranes. The COF membrane formation mechanism is profoundly studied and discussed. However, the ion exchange capacity tests and discussions are not profound enough. The reported results are also similar to their previous works despite the high proton conductivity. Overall, the work may not satisfy the high quality of Nature Communication. The detailed comments are listed below.

1. The increase of n-octanoic acid amount results in raised membrane thicknesses, leading to decreased proton conductivity. From Figure 3b, it can be seen that the membrane crystallinity is strongly influenced by the amount of n-octanoic acid. Thus, the crystallinity of these membranes may affect the proton conductivity as well.

Reply:

Thanks for the reviewer's valuable guidance. We have added some discussion about the effect of crystallinity on the proton conductivity of COF membranes in the Results and Discussion section.

“...As the n-octanoic acid addition amount increased from 5 to 15 mL, the 100 characteristic peak intensity was enhanced (Figure 3b) and the proton conductivity of the membrane was increased from 0.33 S cm⁻¹ to 0.66 S cm⁻¹. The enhanced proton conductivity was attributed to the ordered and continuous ionic channels formed in high crystallinity iCOFMs, which can effectively facilitate the proton transport³...”

2. The synthesized TpPa-SO₃H membranes show lower proton conductivity than TpBD-(SO₃H)₂. Please explain this result.

Reply:

Thanks for the reviewer's valuable guidance. TpBD-(SO₃H)₂ membranes have a larger ion-exchange content (IEC) but a similar swelling ratio to TpPa-SO₃H membranes (Table S7), implying a higher density of sulfonic acid groups in the TpBD-(SO₃H)₂ membranes. Therefore, protons have a shorter hop distance via surface transport mechanism owing to high density and monodispersed sulfonic acid groups (*Adv. Mater.* 2010, 22, 4667; *J. Electroanal. Chem.* 2001, 502, 1). Moreover, owing to the higher IEC, the TpBD-(SO₃H)₂ membranes could absorb more water molecules (Figure S7 and S25). The established models suggest that water molecules play a critical role in proton transport based on Grotthuss and vehicle mechanisms (*Chem. Rev.* 2004, 104, 4637). Therefore, the TpBD-(SO₃H)₂ membranes could achieve higher proton conductivity.

The relevant discussion was added in the revised manuscript. “...The proton conductivity of the resultant TpPa-SO₃H iCOFMs reached 0.2 S cm⁻¹ at 90 °C, 100% RH (Figure S26, Supplementary Information), lower than that of the TpBD-(SO₃H)₂ iCOFMs. This was because TpBD-(SO₃H)₂ membranes have a larger ion-exchange content (IEC) but a similar swelling ratio to TpPa-SO₃H

membranes (Table S7, Supplementary Information), implying a higher density of monodispersed sulfonic acid groups in the TpBD-(SO₃H)₂ membranes. Therefore, protons have a shorter hop distance via surface transport mechanism^{44,45}. Moreover, owing to the higher IEC, the TpBD-(SO₃H)₂ membranes could absorb more water molecules (Figure S7 and S25, Supplementary Information), which play a critical role in proton transport based on Grotthuss and vehicle mechanisms^{46...}

Table S7. Swelling ratio and water uptake of iCOFMs and membranes reported in literature.

Membranes	IEC (mmol g ⁻¹)	Swelling ratio (%)		Temperature (°C)	Conditions	Water uptake (%)	Ref.
		Thickness	Area				
TpBD-(SO ₃ H) ₂	4.6	22	21	25	DI water	144.4	This work
TpBD-(SO ₃ H) ₂		25	23	80			
TpPa-SO ₃ H	3.2	11	14	25		75.3	This work
TpPa-SO ₃ H		15	16	80			
IPC-COF membrane	3.2	9	1	25		60	8
Random copolymer	3.28	110	33	80		183.3	9
	3.19	77	38	80		119	9
	3.26	160	132	50		44	10
	2.51	95	180	50		35	10
Block polymer	1.83	70	25	20		91.2	11
	1.50	65	11	30		48	12
	1.67	55	16	25		82	13
	1.83	10		20	25	14	
Nafion 212®	0.93	14	14	25	12	8	
	0.93	21	44	80	36		

Figure S7. Vacuum water vapor sorption isotherm of TpBD-(SO₃H)₂ membrane.

Figure S25. Vacuum water vapor sorption isotherm of TpPa-SO₃H membrane.

3. The dual-activation polymerization is illustrated in this manuscript. Sodium formate and n-octanoic acid are selected as the activator pairs. The selection criteria should be discussed to inspire others. Are there any other viable activators?

Reply:

Thanks for the reviewer's valuable guidance. To explore the selection criteria of activators, we further selected a variety of amine monomer activators (sodium hydroxide, sodium acetate and sodium benzoate) and aldehyde monomer activators (acetic acid, n-heptanoic acid, n-nonanoic acid and n-decanoic acid) for iCOFMs fabrication.

As shown in Figure S12, we cannot obtain membranes when using a stronger acid (acetic acid) as the aldehyde monomer activator. This was because the acetic acid may diffuse into water and the base cannot sufficiently activate the amine monomer in the water phase. Moreover, we cannot obtain membranes if the amine monomer activator was a strong base (sodium hydroxide), since the strong base would immediately neutralize the acid at the interface.

When using other amine monomer activators and aldehyde monomer activators, a series of iCOFMs can be obtained with different thickness and crystallinity. We found that the membrane thickness increased with the increase of the acidity of the aldehyde monomer activator (Figure S13). This phenomenon can be explained by the membrane growth mechanism. As the acidity of the aldehyde monomer activator increased, aldehyde monomers were more easily to be activated and the initial interface polymerization rate became higher, leading to increased thickness of the loose layer of iCOFMs. Similarly, as the basicity of the amine monomer activator increased, the membrane thickness increased. In addition, the crystallinity of the iCOFMs can also be manipulated by varying the activators (Figure S14).

Therefore, we can obtain the selection criteria of the activators. The aldehyde monomer activator needs to be immiscible with water to form a stable interface and avoid diffusing into water phase. The basicity of the amine monomer activator needs to be strong enough to extract protons from the ionic amine monomer, but should not be too strong to avoid neutralization with the acid in the

organic phase. The addition amount as well as the acidity (basicity) of the activator can be adjusted according to the monomer reactivity as well as required membrane thickness and crystallinity.

The relevant discussion was added in the revised manuscript and Supplementary Information. “...According to the proposed dual activation interfacial polymerization mechanism, other similar Brønsted acids and bases also could be used for monomer activation. To explore the selection criteria of activators, we further selected a variety of amine monomer activators (sodium acetate and sodium benzoate) and aldehyde monomer activators (n-heptanoic acid, n-nonanoic acid and n-decanoic acid) for iCOFMs fabrication. As shown in Figure S13-14 (Supplementary Information), a series of iCOFMs can be obtained with different thickness and crystallinity. We found that the membrane thickness increased with the increase of the acidity of the aldehyde monomer activator. This phenomenon can be explained by the membrane growth mechanism. As the acidity of the aldehyde monomer activator increased, aldehyde monomers were more easily to be activated and the initial interface polymerization rate became higher, leading to increased thickness of the loose layer of iCOFMs. Similarly, as the basicity of the amine monomer activator increased, the membrane thickness increased. In addition, the iCOFMs obtained by coupling sodium formate and n-octanoic acid exhibited highest crystallinity (Figure S14 Supplementary Information). Therefore, sodium formate and n-octanoic acid were selected as model activators for further study...”

“Discussion S3

As shown in Figure S12, we cannot obtain membranes when using a stronger acid (acetic acid) as the aldehyde monomer activator. This was because the acetic acid may diffuse into water and the base cannot sufficiently activate the amine monomer in the water phase. Moreover, we cannot obtain membranes if the amine monomer activator was a strong base (sodium hydroxide), since the strong base would immediately neutralize the acid at the interface. Therefore, we can obtain the selection criteria of the activators. The aldehyde monomer activator needs to be immiscible with water to form a stable interface and avoid diffusing into water phase. The basicity of the amine monomer activator needs to be strong enough to extract protons from the ionic amine monomer, but should not be too strong to avoid neutralization with the acid in the organic phase. The addition amount as well as the acidity (basicity) of the activator can be adjusted according to the monomer reactivity as well as required membrane thickness and crystallinity.”

Figure S12. Fabrication of the iCOFMs using IP technology with (a) stronger acid (acetic acid) and (b) strong base (sodium hydroxide).

Figure S13. Digital photo of TpBD-(SO₃H)₂ membrane (Fabrication condition: a. 15 mL n-octanoic acid for aldehyde monomer activation and 2.0 eq sodium formate/ sodium acetate/ sodium benzoate for amine monomer activation; b. 15 mL n-heptanoic acid/ n-octanoic acid/ n-nonanoic acid/ n-decanoic acid for aldehyde monomer activation and 2.0 eq sodium acetate for amine monomer activation).

Figure S14. PXRD patterns of TpBD-(SO₃H)₂ membrane (Fabrication condition: a. 15 mL n-octanoic acid for aldehyde monomer activation and 2.0 eq sodium formate/ sodium acetate/ sodium benzoate for amine monomer activation; b. 15 mL n-heptanoic acid/ n-octanoic acid/ n-nonanoic acid/ n-decanoic acid for aldehyde monomer activation and 2.0 eq sodium acetate for amine monomer activation).

4. The weight loss in 40-150°C is ~85%. This is very considerable, which may not arise from the evaporation of absorbed water.

Reply:

Thanks for the reviewer's valuable guidance. To analyze the reason for about 15% weight loss in 40-150 °C, we further carried out a vacuum water vapor sorption test on TpBD-(SO₃H)₂ and TpPa-SO₃H membrane. It was found that at 30 °C and 50% relative humidity, the water vapor adsorption capacity was 229-247 mg g⁻¹ (Figure S7 and S25). The high water adsorption was due to the high specific area and abundant hydrophilic -SO₃H groups within the membrane pores. As the membrane sample was kept under the conditions of 25 °C and 53 RH% and no special dehydration treatment was carried out before the TGA test, we derive that about 15% of the weight loss at 40-150 °C on the TGA test could be attributed to the evaporation of adsorbed water.

Figure S7. Vacuum water vapor sorption isotherm of TpBD-(SO₃H)₂ membrane.

Figure S25. Vacuum water vapor sorption isotherm of TpPa-SO₃H membrane.

5. It is interesting to find that thus-prepared iCOF membranes hold higher crystallinity than their bulk powders synthesized under solvothermal conditions. It will be valuable to guide the synthesis of highly crystalline COF membranes if the underlying mechanism is discussed.

Reply:

Thanks for the reviewer's valuable guidance. Previous studies have proved that suitable reaction rate, reversibility and low growth rate are reliable conditions for the synthesis of high crystallinity COFs (*Science*, 2018, 361, 48; *Angew. Chem. Int. Ed.*, 2019, 58, 2; *J. Am. Chem. Soc.*, 2019, 141, 1807). On one hand, the solvothermal reaction system for COF powder synthesis was homogeneous. We can only activate one type of monomer by adding either aldehyde monomer activator (acid) or

amine monomer activator (base). On the other hand, during the homogeneous solvothermal synthesis for COF powder, the two monomers directly contact and polymerize rapidly at high temperatures. During the interfacial polymerization process for the fabrication of iCOFMs, the growth rate was limited by the diffusion rate of the monomers across the interface, which can maintain efficient self-correction for higher crystallinity.

The relevant discussion was added in the revised Supplementary Information.

“Discussion S1

Previous studies have proved that suitable reaction rate and reversibility and low growth rate are reliable conditions for the synthesis of high crystallinity COFs¹⁻³. On one hand, the solvothermal reaction system for COF powder synthesis was homogeneous. We can only activate one type of monomer by adding either aldehyde monomer activator (acid) or amine monomer activator (base). On the other hand, during the homogeneous solvothermal synthesis for COF powder, the two monomers directly contact and polymerize rapidly at high temperatures. During the interfacial polymerization process for the fabrication of iCOFMs, the growth rate was limited by the diffusion rate of the monomers across the interface, which can maintain efficient self-correction for higher crystallinity.”

6. Given the asymmetric structure of membranes, i.e. top dense films and loosely stacked particles, the membrane may contain a crystallinity asymmetry. The crystallinity of the film and particles should be presented separately.

Reply:

Thanks for the reviewer’s valuable guidance. We used grazing incidence XRD (GIXRD) and GIWAXS to characterize the crystallinity of the top dense side and loosely stacked side of the COF membranes (Figure S5-6). It was found that the crystallinity of the top dense side of the membrane was slightly higher than that of the loosely stacked side.

This is because the crystallinity of COFs was significantly affected by the growth rate (*Science*, 2018, 361, 48). In this study, when the interfacial polymerization started, the two-phase monomers reacted rapidly, and the faster growth rate could result in COF particles with relatively low crystallinity (*Sci. Adv.* 2020, 6, eabb1110). As the reaction time prolonged, the COF particles will stack into a loose layer, which reduced the further diffusion rate of amine monomers into the organic phase. This is similar to the self-sealing and self-terminating behavior in the interfacial polymerization process for polyamide membranes (*J. Mater. Chem. A*, 2019, 7, 25641; *J. Mater. Chem. A*, 2020, 8, 23930). Accordingly, the interfacial polymerization was slowed down, leading to the formation of a compact layer on top side. The slower growth rate can maintain efficient self-correction of defects, leading to higher crystallinity of the top dense layer (*J. Am. Chem. Soc.* 2018, 140, 5145, *Sci. Adv.* 2020, 6, eabb1110).

The relevant discussion was added in the revised Manuscript and Supplementary Information.

“...Meanwhile, we conducted the grazing incidence wide-angle X-ray scattering (GIWAXS) and grazing incidence XRD (GIXRD) measurement to analyze the crystallinity of the membrane surface. It was found that the crystallinity of the top dense side of the membrane was slightly higher than that of the loosely stacked side (Figure 1e and Figure S5-6, Supplementary Information)...”

“Discussion S2

It was found that the crystallinity of the top dense side of the membrane was slightly higher than that of the loosely stacked side. This is because the crystallinity of COFs was significantly affected by the growth rate¹. In this study, when the interfacial polymerization started, the two-phase monomers reacted rapidly, and the faster growth rate could result in COF particles with relatively low crystallinity⁴. As the reaction time prolonged, the COF particles will stack into a loose layer, which reduced the further diffusion rate of amine monomers into the organic phase. This is similar to the self-sealing and self-terminating behavior in the interfacial polymerization process for polyamide membranes^{5,6}. Accordingly, the interfacial polymerization was slowed down, leading to the formation of a compact layer on top side. The slower growth rate can maintain efficient self-correction of defects, leading to higher crystallinity of the top dense layer^{4,7}.”

Figure S5. GIXRD pattern of (a) top dense side and (b) loosely stacked side of the TpBD-(SO₃H)₂ membrane.

Figure S6. GIWAXS pattern of (a) top dense side and (b) loosely stacked side of the TpBD-(SO₃H)₂ membranes.

7. The sulfonic acid groups can be easily introduced into COF skeletons by post-synthetic strategies. The advantage of this De Novo method is suggested to be added.

Reply:

Thanks for the reviewer's valuable guidance. For post-synthetic strategies, the COF skeletons still need to possess active sites, such as -OH and -CN (*Angew. Chem. Int. Ed.* 2019, 58, 2; *J. Am. Chem. Soc.* 2015, 137, 7079; *Nat. Chem.* 2019, 11, 587), which can further react with functional modifiers. Moreover, the extrinsic functional modifiers usually need to diffuse into the nanosized pores of COFs for post-modification, which is challenging to control. In this study, the pore size of the COF membrane is below 3 nm, while the membrane thickness is 3 μm. The accessibility of extrinsic functional modifiers was quite limited, and it is difficult to guarantee the all-dimensional and homogeneous modification of all reaction sites inside the COF membrane. Compared to post-synthetic strategies, *De Novo* strategies by designing ionic monomers can guarantee the all-dimensional and homogeneous modification of all reaction sites inside the COF membrane and has been proved effective for iCOFMs design (*Adv. Mater.*, 2020, 32, 2001284; *Adv. Mater.*, 2020, 32, 2005565; *Angew. Chem. Int. Edit.*, 2021, 60, 17638).

The relevant discussion was added in the Introduction section of revised manuscript. "...Increasing ion exchange capacity (IEC) of membranes by introducing multiple ion groups into skeletons was the key to push the upper limit of iCOFM proton conductivity^{3,10}. Compared to post-synthetic strategies, *De Novo* strategies by designing ionic monomers can guarantee the all-dimensional and homogeneous modification of all reaction sites inside the COF membrane and has been proved effective for iCOFMs design^{3,4,17}..."

8. The HRTEM image in Figure 1f is not clear, please enhance the resolution.

Reply:

Following the reviewer's valuable guidance, we have re-taken the HRTEM image of the iCOFMs section and acquired high resolution HRTEM image Figure 1f.

Figure 1. (f) High-resolution TEM images of the slices of iCOFMs

9. There are some grammatical errors that need revision.

Reply:

Thanks for the reviewer's valuable guidance. We have carefully checked and corrected the grammatical errors in the revised manuscript. Some typical corrections are shown below.

“Benefited from the simplicity and scalability, interfacial polymerization (IP) has evolved as a platform technology for COF membrane fabrication by confining the polymerization reactions between monomers in two immiscible phases at the interface¹⁹, during which the monomer reactivity directly governs the membrane structure formation^{24,31}.” was revised as “Benefited from the simplicity and scalability, interfacial polymerization (IP) has evolved as a platform technology for COF membrane fabrication by confining the polymerization reactions between monomers in two immiscible phases at the interface¹⁹. During IP process, the monomer reactivity directly governs the membrane structure formation^{24,31}.”

“However, the IP technology for fabricating nonionic COF membranes, can not directly transplant into iCOFMs fabrication.” was revised as “However, the IP technology for fabricating non-ionic COF membranes can hardly be transplanted into iCOFMs fabrication directly.”

“...indicating that the reactivity of amine monomer was decreased as the number of ionic groups increases.” was revised as “indicating that the reactivity of amine monomer was decreased as the number of ionic groups increased.”

“...and COF nanosheets formed in the water phase (Figure 2b).” was revised as “...and COF nanosheets were formed in the water phase (Figure 2b).”

“Under the dual-activation condition (Figure 2d), an iCOFMs formed with a thickness of 85 μm was formed,...” was revised as “Under the dual-activation condition (Figure 2d), an iCOFMs with a thickness of 85 μm was formed,...”

“...the initial interface polymerization rate became faster, lead to the formation of more non-continuous COF particles...” was revised as “...the initial interface polymerization rate became higher, leading to the formation of more non-continuous COF particles...”

10. The iCOF membranes give higher proton conductivities than others and commercial materials. How about the stability of proton conductivity?

Reply:

Following the reviewer's valuable guidance, we have carried out the long-term operation test of membrane proton conductivity at 90 °C and 100% RH. The results were added in Figure S27 in the revised manuscript, as shown below. During the test, and the proton conductivity of the TpBD-(SO₃H)₂ membrane only slightly decreased by about 7.6% within the first 5 hours, which was probably due to the instability of the amorphous regions of the membrane under high temperature and high humidity, and then remained stable until two weeks, verifying the membrane's excellent long-term thermal durability for further fuel cell applications.

The relevant discussion was added in the revised manuscript. "...Moreover, we have carried out the long-term operation test of membrane proton conductivity at 90 °C and 100% RH (Figure S27). During the test, and the proton conductivity of the TpBD-(SO₃H)₂ membrane only slightly decreased by about 7.6% within the first 5 hours, which was probably due to the instability of the amorphous regions of the membrane under high temperature and high humidity, and then remained stable until two weeks, verifying the membrane's excellent long-term thermal durability for further fuel cell applications..."

Figure S27. long-term proton conductivity of the TpBD-(SO₃H)₂ membrane

REVIEWER COMMENTS

Reviewer #1 (Remarks to the Author):

The authors did a proper revision. Acceptance of the manuscript is recommended.

2 minor remarks:

- On pages 22/23 and maybe in the manuscript, there is a mix of "Fukui Function" and "Fukui function".
- On pages 21/23, some symbols are not visible.

Reviewer #2 (Remarks to the Author):

The authors have addressed part of the concerns, but they did not provide sufficient results to answer the raised questions 3, 5, 6, and 7. Detailed comments are given as follows.

3. The authors have showed the re-tested HRTEM images in Figure 1f. However, the TEM and EDS images were not updated.

5. The authors have re-tested the iCOFMs for three times, and found only one spike in the re-tested XRD pattern. The peak at 2.6 degree in the previous test may be caused by sample contamination. But there are four samples in the original manuscript that all show double peaks, all of which were all caused by sample contamination?

6. The authors have carried out the long-term operation test of membrane proton conductivity at 90 °C and 100% RH. The proton conductivity of the iCOFMs decreased by about 7.6% within the first 5 hours, and then remained stable within two weeks. Authors should discuss these two values separately in the manuscript, while not only mention the highest value.

7. Please check carefully the writing of T_p , e.g. in Figure 2e.

Reviewer #3 (Remarks to the Author):

The authors have made an excellent revision and the manuscript is acceptable for publication.

Reviewer #1 (Remarks to the Author):

The authors did a proper revision. Acceptance of the manuscript is recommended.

2 minor remarks:

- On pages 22/23 and maybe in the manuscript, there is a mix of "Fukui Function" and "Fukui function".

Reply:

Following this reviewer's highly positive remarks and valuable guidance, we have checked and revised the "Fukui Function" as "Fukui function" in the revised manuscript, as shown below.

Page 10

"...As shown in Figure 2e-f, without activation, the Fukui function f^+ for nucleophilic attack sites on the aldehyde monomer was 0.066..."

Page 22

"...In this study, Fukui function was calculated for monomers reactivity discussed..."

"...Fukui function of monomers was shown in Tables S2-6 (Supplementary Information)..."

- On pages 21/23, some symbols are not visible.

Reply:

Following this reviewer's valuable guidance, we have checked the written errors of symbols in pages 21-23, as shown below.

"...The mixture was sonicated for 20 min and frozen in liquid nitrogen, followed by degassing treatment by three freeze-pump-thaw cycles. Then, the tube was sealed and heated at 120 °C for 3 days. The deep red powder was collected by filtration, washed with anhydrous ethanol, anhydrous acetone and deionized water for 3 times and subjected to Soxhlet extraction with methanol for 3 days. Finally, the deep red powder was collected after drying at 120 °C under vacuum. Yield: 107.8 mg, 74.2%..."

"...GIWAXS measurement was performed by a Mars345 detector with X-ray wavelength of 0.15418 nm, the incident angle was 0.5° and the distance of sample to detector was 438 mm. AFM (Dimension Icon, Bruker Co. Ltd. USA) analysis was utilized to characterize the surface roughness of the membrane. The stress-strain curves of the membranes were obtained by a Yangzhou Zhongke WDW-02 electronic stretching machine with a strain rate of 2 mm min⁻¹ under ambient condition. Thermogravimetric analysis (TGA) was carried out on a NETZSCH 209F3 thermal analyzer at a heating rate of 10 °C min⁻¹ from 40 to 800 °C and under N₂ atmosphere..."

“...The measurements were conducted at temperatures between 30 °C and 90 °C inside a thermohygrostat. The proton conductivities (σ , S cm⁻¹) were calculated as follows...”

Reviewer #2 (Remarks to the Author):

The authors have addressed part of the concerns, but they did not provide sufficient results to answer the raised questions 3, 5, 6, and 7. Detailed comments are given as follows.

3. The authors have showed the re-tested HRTEM images in Figure 1f. However, the TEM and EDS images were not updated.

Reply: Thanks for the reviewer's valuable guidance. We have updated the TEM and EDS images in Figure 1f, as shown below.

Figure 1. (f) TEM, EDS, and high-resolution TEM images of the slices of iCOFMs.

5. The authors have re-tested the iCOFMs for three times, and found only one spike in the re-tested XRD pattern. The peak at 2.6 degree in the previous test may be caused by sample contamination. But there are four samples in the original manuscript that all show double peaks, all of which were all caused by sample contamination?

Reply:

Thanks for the reviewer's valuable guidance. We re-prepared four kinds of iCOFMs in Figure 3b and 3e. The digital photos were shown in Figure R1. Three copies of each kind of membrane were prepared and performed for XRD tests. As shown in Figure R2, all four kinds of membranes show only one spike at 3.4 degree and no peak at 2.6 degree. Therefore, we can guarantee the validity and reproducibility of the XRD data in Figure 3b and 3e of the revised manuscript.

In addition, we tried to find the reason of the appearance of the peaks at 2.6 degree in the original manuscript. As all samples were rinsed with DMF and ethyl alcohol, and were carefully kept before tests, we suspect that the peaks at 2.6 degree shown in Figure 3b and 3e in the original manuscript may be caused by contamination during the XRD test process. As shown in Figure R3, during XRD

test, the membrane sample was directly placed on a sample stage, which is a silicon wafer specially made by Rigaku Corporation and has no diffraction peak in the range of 2-120 degrees. Under normal wide-angle XRD test conditions, the X-ray detection depth could exceed 100 μm for low-density organic and porous materials (*J. Pharm. Sci.* 2010, 99, 3807-3814). If some contaminants existed on the sample stage, characteristic peaks of the contaminants could appear in the XRD pattern.

Moreover, all four samples, which exhibited diffraction peaks at 2.6 degree in the original manuscript, were tested sequentially using the same sample stage at one time. Therefore, we suspect that the sample stage we used may have been contaminated, probably due to the residuals from previous tests.

Figure R1. Digital photo of the four kinds of TpBD-(SO₃H)₂ iCOFMs (Fabrication condition: a. 20 mL n-octanoic acid for aldehyde monomer activation and 2.0 eq sodium formate for amine monomer activation; b. 15 mL n-octanoic acid for aldehyde monomer activation and 2.5 eq sodium formate for amine monomer activation; c. 15 mL n-octanoic acid for aldehyde monomer activation and 2.0 eq sodium formate for amine monomer activation; d. 15 mL n-octanoic acid for aldehyde monomer activation and 1.5 eq sodium formate for amine monomer activation).

Figure R2. PXRD patterns of the four kinds of TpBD-(SO₃H)₂ iCOFMs (Fabrication condition: a. 20 mL n-octanoic acid for aldehyde monomer activation and 2.0 eq sodium formate for amine monomer activation; b. 15 mL n-octanoic acid for aldehyde monomer activation and 2.5 eq sodium formate for amine monomer activation; c. 15 mL n-octanoic acid for aldehyde monomer activation and 2.0 eq sodium formate for amine monomer activation; d. 15 mL n-octanoic acid for aldehyde monomer activation and 1.5 eq sodium formate for amine monomer activation; Three copies of each kind of membrane were prepared and performed for XRD tests).

Figure R3. XRD sample stage before and after putting membrane sample.

6. The authors have carried out the long-term operation test of membrane proton conductivity at 90 °C and 100% RH. The proton conductivity of the iCOFMs decreased by about 7.6% within the first 5 hours, and then remained stable within two weeks. Authors should discuss these two values separately in the manuscript, while not only mention the highest value.

Reply:

Following the reviewer's valuable guidance, we have added some discussions about long-term operation proton conductivity values in the Results and discussion section as well as in the Conclusion section of the revised manuscript.

"...During the test, and the proton conductivity of the TpBD-(SO₃H)₂ iCOFMs only slightly decreased by about 6% to 0.62 S cm⁻¹ within the first 6 hours, which was probably due to the instability of the amorphous regions of the membrane under high temperature and high humidity. The proton conductivity of TpBD-(SO₃H)₂ iCOFMs remained stable around 0.62 S cm⁻¹ within two weeks, verifying the membrane's excellent long-term thermal durability for further fuel cell applications..."

"...outperforming all the currently reported COF-based proton-conducting materials. Moreover, TpBD-(SO₃H)₂ iCOFMs also show excellent proton conductivity around 0.62 S cm⁻¹ during the two-week long-term proton conductivity test (90 °C, 100% relative humidity). We anticipate that our membranes can find diverse applications such as fuel cell, flow cell, lithium battery and nanofiltration..."

7. Please check carefully the writing of Tp, e.g. in Figure 2e.

Reply:

Thanks for the reviewer's valuable guidance. We have checked the written errors and unified the abbreviation for 2,4,6-triformylphloroglucinol as Tp in the revised manuscript, as shown below.

Page 11

"...The Fukui function f^+ of carbonyl carbon atom on Tp molecule increased to 0.152, indicating the increased monomer reactivity. For amine monomers, it can be seen from the optimized BD-(SO₃H)₂ structure..."

Page 19-21

"...2,4,6-triformylphloroglucinol (Tp) was bought from Jilin Chinese Academy of Sciences - Yanshen Technology Co., Ltd., China..."

"...0.2 mmol of 2,4,6-triformylphloroglucinol (Tp) was dissolved into a mixed solvent of n-octanoic acid and mesitylene..."

“...The amine monomer solution was poured into the bottom of 100 mL beaker, and the **Tp** solution was added by drops on the top layer...”

“...Typically, 0.2 mmol (42.0 mg) of 2,4,6-triformylphloroglucinol (**Tp**), 0.3 mmol (103.3mg) of 4,4'-diaminobiphenyl-3,3'-disulphonic acid (BD-(SO₃H)₂)...”

Figure 2

Figure 2. (e) Fukui functions for nucleophilic attack sites (f^+) of aldehyde monomer.

Supplementary Information

“...0.2 mmol (42.0 mg) of 2,4,6-triformylphloroglucinol (**Tp**) was dissolved into 20 mL n-octanoic acid by sonication for 30 min...”

“...The amine monomer solution was poured into the bottom of 100 mL beaker, and the **Tp** solution was added by drops on the top layer ...”

“...Typically, 0.2 mmol of 2,4,6-triformylphloroglucinol (**Tp**), 0.3 mmol (56.4 mg) of 2,5-diaminobenzenesulfonic acid (Pa-SO₃H), 3 mL mesitylene, 1 mL dioxane and acetic acid (6 mol L⁻¹, 0.5 mL) were added in a 15 mL pyrex tube...”

“...2,4,6-triformylphloroglucinol (**Tp**)...”

Reviewer #3 (Remarks to the Author):

The authors have made an excellent revision and the manuscript is acceptable for publication.

Reply:

Thanks to the reviewer for the highly positive remarks.

REVIEWER COMMENTS

Reviewer #2 (Remarks to the Author):

The authors have satisfyingly addressed the comments. The decision of acceptance is recommended.

Reviewer #2 (Remarks to the Author):

The authors have satisfyingly addressed the comments. The decision of acceptance is recommended.

Reply:

Thank the reviewer for the highly positive comments and the efforts in reviewing this manuscript.